# A Fully First-Order Layer for Differentiable Optimization

**Zihao Zhao** [* 1]   **Kai-Chia Mo** [* 2]   **Shing-Hei Ho** [1]   **Brandon Amos** [3]   **Kai Wang** [1]

## Abstract

Differentiable optimization studies how to embed a mathematical program as a differentiable layer in machine learning pipelines. However, existing approaches typically rely on implicit differentiation, involving expensive Hessian computation while differentiating through optimality conditions. To address this challenge, we formulate the differentiable optimization problem as a bilevel optimization instance. We construct a new active-set Lagrangian as a proxy to compute an $\epsilon$-approximate hypergradient using only near-constant $\mathcal{O}(\log(1/\epsilon))$ first-order information. We also show that applying this efficient hypergradient oracle to constrained bilevel optimization improves the overall gradient complexity to $\tilde{\mathcal{O}}(\delta^{-1}\epsilon^{-3})$ to reach a $(\delta, \epsilon)$-Goldstein stationary point. We implement our method `FFOLayer`, as a drop-in Python library compatible with existing differentiable optimization solvers. Our algorithm shows significantly faster computation with similar convergence compared to other existing solvers. The source code is available at https://github.com/GT-KOALA/FFOLayer.

## 1. Introduction

Modern machine learning pipelines increasingly integrate optimization problems as differentiable layers to enable end-to-end decision-making (Amos & Kolter, 2017; Agrawal et al., 2019a; Bertrand et al., 2022). In this paradigm, the model's output is obtained by solving an embedded optimization problem, and gradients are backpropagated through the solver to train end-to-end. This approach has shown promise in various decision-making applications, such as decision-focused learning (Donti et al., 2017; Wilder et al., 2019; Mandi et al., 2024), control (Amos et al., 2018),

and meta-learning (Lee et al., 2019; Lorraine et al., 2020).

Although widely used, a major bottleneck for differentiable optimization layers is scalability: computing hypergradients can be extremely costly (Bolte et al., 2023). Standard approaches differentiate the Karush–Kuhn–Tucker (KKT) conditions via implicit differentiation (Agrawal et al., 2019a), reducing backpropagation to several linear solves of the KKT system. However, factorizing and solving a large KKT system are both memory- and time-intensive in practice, which limits the problem size (Yang et al., 2021; Petrulionyte et al., 2024). This issue motivates research into more efficient methods for computing hypergradients.

In this paper, we address these challenges by developing a novel algorithm that uses only **first-order information**. Our key insight is that the dependency of the optimization's solution on the inputs can be captured by a *bilevel optimization* structure. By reformulating the differentiable optimization problem as a bilevel problem, we construct an active-set Lagrangian to compute an approximate hypergradient with only first-order oracle calls. This hypergradient approximation avoids costly second-order operations and significantly improves scalability.

On the theoretical side, we show that our proposed algorithm computes an $\epsilon$-approximate hypergradient using only first-order information in $\mathcal{O}(\log(1/\epsilon))$ time, under appropriate conditions. For constrained bilevel problems, this strengthens prior results by twofold. First, we improve the convergence rate of linearly constrained bilevel optimization to $\tilde{\mathcal{O}}(\delta^{-1}\epsilon^{-3})$ to find a $(\delta, \epsilon)$-Goldstein stationary point (Goldstein, 1977), matching the best known rate for nonsmooth nonconvex optimization (Zhang et al., 2020b). Second, we extend the guarantee from the linearly constrained bilevel setting to the general convex-constrained setting, showing that the same $\tilde{\mathcal{O}}(\delta^{-1}\epsilon^{-3})$ rate in oracle complexity under additional assumptions.

We implement our approach as an open-source PyTorch library `FFOLayer` and demonstrate that it can be easily integrated into existing codebases. Notably, our method is *solver-agnostic*: hypergradients are computed without solver-specific differentiation, relying only on black-box solves of the original and perturbed problems. This property allows us to leverage state-of-the-art convex program solvers (such as GUROBI (Gurobi Optimization, LLC, 2025) and

---
[*]Equal contribution  [1]School of Computational Science and Engineering Georgia Institute of Technology, GA, USA [2]Independent researcher. [3]Reflection AI, NY, USA. Correspondence to: Zihao Zhao <zzhao628@gatech.edu>.

*Proceedings of the 43$^{rd}$ International Conference on Machine Learning*, Seoul, South Korea. PMLR 306, 2026. Copyright 2026 by the author(s).

MOSEK (ApS, 2025)) without modifying the gradient computation process, which is not supported by most of the existing differentiable optimization layers.

Our contributions can be summarized as follows:

- We first rewrite the constrained differentiable optimization as a bilevel optimization. We then design a new first-order (inexact) oracle that computes $\epsilon$-approximate hypergradient with $\mathcal{O}(\log(1/\epsilon))$ computation cost for general constrained bilevel optimization.
- We provide theoretical convergence analysis for both linear and general convex constraints, showing that our method matches the best-known convergence complexity $\tilde{\mathcal{O}}(\delta^{-1}\epsilon^{-3})$ for reaching a Goldstein stationary point in constrained bilevel optimization.
- We release an open-source solver-agnostic layer that can be seamlessly integrated into neural network training pipelines, offering a drop-in replacement for existing differentiable solvers.

*Table 1.* Comparison of differentiable convex optimization layers. Solver-agnostic: whether a method can adapt to any state-of-the-art solver. ✓ indicates the method has the property, ✗ indicates it does not. † $\epsilon$-approximate hypergradient.

| Method | Convex constraints | First-order only | Solver-agnostic | Exact hypergradient |
|---|---|---|---|---|
| *KKT-based* | | | | |
| CvxpyLayer | ✓ | ✗ | ✗ | ✓ |
| QPTH | ✗ | ✗ | ✗ | ✓ |
| *Optimization-based* | | | | |
| Alt-Diff | ✗ | ✗ | ✗ | ✓ |
| LPGD | ✓ | ✓ | ✓ | ✗ |
| BPQP | ✓ | ✗ | ✗ | ✓ |
| dQP | ✗ | ✗ | ✓ | ✓ |
| FFOLayer (**ours**) | ✓ | ✓ | ✓ | ✗† |

## 2. Related Works

**Differentiable Optimization** The idea of treating optimization problems as differentiable modules of a neural network dates back to early convex formulations. Amos & Kolter (2017) introduced OptNet, which embeds a quadratic program solver as a layer whose forward pass solves the optimization and whose backward pass recovers the hypergradient by implicitly differentiating the KKT system. Subsequent work accelerated the computation speed of QP (Magoon et al., 2025), supporting infeasible problems (Butler, 2023; Bambade et al., 2024), and scaling-up the problem size (Butler & Kwon, 2023). Beyond QP, some works generalized this paradigm to broader classes of convex programs with toolkit (Agrawal et al., 2019a; Blondel et al., 2022; Ren et al., 2023; Sun et al., 2023; Besançon et al., 2024; Paulus et al., 2024; Pan et al., 2024; Schaller & Boyd, 2025), differentiable conic layers (Agrawal et al., 2019b;

Healey et al., 2025), combinatorial optimization (Paulus et al., 2021), and non-convex problems such as robotics-oriented solvers (Innes et al., 2019; Pineda et al., 2022; Tracy et al., 2023; Holmes et al., 2025; Rosemberg et al., 2025).

Despite these layers having enabled end-to-end learning, they also expose the computational burden of differentiating through structured optimization problems. Table 1 summarizes prior methods aimed at reducing differentiation cost. Notably, while CvxpyLayer (Agrawal et al., 2019a) supports general convex constraints, it is not fully first-order. QPTH (OptNet) (Amos & Kolter, 2017) supports only solving quadratic programs (QPs). Alt-Diff (Sun et al., 2023) computes exact hypergradients via alternative differentiation, but it does not support general convex constraints and is not first-order. LPGD (Paulus et al., 2024) is a first-order solver-agnostic method, but only ensures asymptotic guarantees. BPQP (Pan et al., 2024) / dQP (Magoon et al., 2025) transforms a general convex program/QP into QP form to leverage efficient QP solvers, avoiding solving the Hessian inverse but adding overhead from problem transformation. In contrast, FFOLayer is the first method to combine all these properties: it handles general convex constraints, uses only first-order information, can be equipped with all advanced solvers, and comes with finite-time guarantees for hypergradient approximation and convergence to a stationary point.

**Bilevel Optimization with First-Order Algorithms** When the differentiable layer is formulated as a bilevel problem, computing hypergradients traditionally requires explicit Hessian or Jacobian-vector products, which are often the bottleneck in large-scale applications. This has motivated a wave of first-order bilevel algorithms that avoid Hessians entirely. Value-function (VF) approaches (Kwon et al., 2023; 2024a; Kornowski et al., 2024) replace the lower-level problem with its value function and differentiate through a penalty reformulation, requiring only first-order oracle calls. This technique extends naturally to constrained lower-level objectives (Kornowski et al., 2024; Zhang et al., 2024; Jiang et al., 2024; Yao et al., 2024) and can attain nearly optimal complexity guarantees in the strongly-convex or Polyak-Łojasiewicz regime (Shen & Chen, 2023; Xiao et al., 2023; Sun et al., 2023).

The key bottleneck now for constrained bilevel optimization is how the inequality constraints are handled: existing works typically enforce inequalities via *penalty formulations* (such as $\ell_2$ norm of active constraints), so obtaining an $\epsilon$-accurate hypergradient in $\mathcal{O}(1/\epsilon)$ time, whereas equality-only formulations admit much faster $\mathcal{O}(\log(1/\epsilon))$ hypergradient approximation (Kornowski et al., 2024). Our work aims to close this gap for differentiable optimization layers by introducing an active-set Lagrangian oracle that reduces

general convex constraints locally to a trackable equality-constrained surrogate, and thus obtains $\mathcal{O}(\log(1/\epsilon))$ hypergradient approximation. Furthermore, we extend these guarantees to the "well-behaved" general convex constraints highlighted as an open problem by Kornowski et al. (2024).

## 3. Problem Statement

We study learning with differentiable convex optimization layers. Let $x \in \mathbb{R}^m$ denote the layer parameter (e.g., a neural network), and let the downstream loss be $f : \mathbb{R}^m \times \mathbb{R}^d \to \mathbb{R}$. We define the training objective $F(x) := f(x, y^*(x))$, where $y^*(x)$ is the solution returned by solving the optimization layer parameterized by $x$. Formally, for each $x$, we define

$$y^* \in \arg \min_{y \in \mathbb{R}^d} g(x, y) \ \text{s.t.} \ h(x, y) \leq 0, e(x, y) = 0, \quad \text{(P0)}$$

where $g(x, y)$ is a convex objective function, $h(x, y) \leq 0$ represents a set of convex inequality constraints, and $e(x, y)$ represents affine equality constraints.

During end-to-end training, we need to compute the *hypergradient* $\nabla_x F(x)$. By chain rule, the hypergradient can be expanded as follows:

$$\nabla_x F = \nabla_x f(x, y^*(x)) + (\frac{dy^*(x)}{dx})^\top \nabla_y f(x, y^*). \quad (1)$$

The main computational challenge is to efficiently compute the sensitivity term $dy^*(x)/dx$.

A common approach is to employ the implicit function theorem (Dontchev & Rockafellar, 2009; Amos & Kolter, 2017), which can be regarded as solving a KKT system. At optimum, there exist Lagrange multipliers $(\lambda, \nu)$ for the inequality and equality constraints such that the primal-dual pair $(y^*, \lambda^*, \nu^*)$ satisfies the KKT conditions. We define the KKT system for Problem P0 as

$$G := \begin{bmatrix} \nabla_y g(x, y) + \nabla_y h(x, y)^\top \lambda + \nabla_y e(x, y)^\top \nu \\ \lambda \circ h(x, y) \\ e(x, y) \end{bmatrix}.$$

Plugging in the optimum, the KKT system satisfies $G(y^*, \lambda^*, \nu^*, x) = 0$. Since $(y^*, \lambda^*, \nu^*)$ implicitly depends on $x$, we can apply the chain rule to compute their total derivative with respect to $x$ and set the derivative to 0 to get the *implicit differentiation formula*:

$$[dy^*, d\lambda^*, d\nu^*]/dx = -(\nabla_{(y,\lambda,\nu)}G)^{-1}\nabla_x G, \quad (2)$$

Importantly, the term $dy^*(x)/dx$ requires solving a linear system involving the inverse of the Jacobian $(\nabla_{(y,\lambda,\nu)}G)^{-1}$, which can be computationally expensive and memory-intensive to compute for large problems. Besides, in most problems, the $\nabla_{(y,\lambda,\nu)}G$ involves computing a Hessian

$\nabla_{yy}^2 g(x, y)$, which poses a major bottleneck for scaling differentiable optimization layers. These challenges motivate our goal of approximating the gradient $\nabla_x F$ using only first-order information, thereby eliminating the need to invert or even form a large KKT matrix.

## 4. Bilevel Reformulation with Active-set Reduction

In this section, we provide an alternative formulation of differentiable optimization problems that avoids explicit Hessian or inverse-Jacobian formalization. Our approach rewrites the differentiable optimization as a bilevel problem, enabling the computation of hypergradients using only first-order information.

### 4.1. Bilevel Formalization for Differentiable Optimization

Following the notation of Section 3, we can express the differentiable optimization as the following bilevel instance:

$$\min_x F(x) := f(x, y^*)$$
$$\text{s.t.} \ y^* \in \arg\min_{y:h(x,y)\leq 0, e(x,y)=0} g(x, y). \quad \text{(P1)}$$

Here, $f(x, y)$ is the *upper-level* objective, $g(x, y)$ is the *lower-level* objective, and $h(x, y), e(x, y)$ are the lower-level constraints. As established in Section 3, the gradient of $\nabla_x F$ can be obtained by implicitly differentiating the lower-level optimality conditions as Equation 2.

After we formalize the problem as a bilevel optimization, we can resort to bilevel algorithms to solve it. We next specify the stationary notion we target and list some standard assumptions in bilevel literature.

**Definition 4.1** (Goldstein stationary point). Let $f : \mathbb{R}^d \to \mathbb{R}$ be Lipschitz and $x \in \mathbb{R}^d$. We say $x$ is $(\delta, \epsilon)$-stationary if $\text{dist}(0, \partial f(x + \delta B)) \leq \epsilon$, where $B$ is a unit ball and $\text{dist}(x, S) := \inf_{y \in S} \|x - y\|$.

**Assumption 4.2.** We assume

1. Upper-level (UL): The objective $f$ is $C_f$-smooth and $L_f$-Lipschitz continuous in $(x, y)$.
2. Lower-level (LL): The objective $g$ is $C_g$-smooth. Fixing any $x \in \mathcal{X}$, $g(x, \cdot)$ is $\mu_g$-strongly convex.
3. We assume that the linear independence constraint qualification (LICQ) condition holds for the LL problem at every $x$ and its corresponding primal solution $y^*$, i.e., the Jacobian of the (active) LL constraints with respect to $y$ at $(x, y^*(x))$ has full row rank.

**Assumption 4.3** (Active-set identification). In solving Problem P1, the active constraints can be correctly identified.

*Remark* 4.4. Although existing work has used Assumption 4.3 (see, for example, Remark 1 in (Khanduri et al.,

2025)), we revisit it in Appendix F, where we clarify why it is needed and establish a weaker guarantee that holds without this assumption.

A main challenge in bilevel optimization is the efficient evaluation of the hypergradient. This motivates leveraging recent first-order bilevel techniques to efficiently compute (or estimate) $\nabla_x F(x)$. For instance, for linear *equality-constrained* lower-level problems, Kornowski et al. (2024) show that an $\epsilon$-approximate hypergradient can be computed in $\mathcal{O}(\log(1/\epsilon))$ time. However, handling *inequality* constraints is more challenging: in the method of Kornowski et al. (2024), inequalities are enforced via a penalty term, and achieving an $\epsilon$-accurate hypergradient requires $\mathcal{O}(\epsilon^{-1})$ iterations, incurring a significantly higher cost than the equality-only case. The gap motivates a more direct approach to handling active constraints.

### 4.2. Active-Set Bilevel Optimization Reduction

We address the aforementioned challenge for **general convex-constrained** problems beyond only linear inequality constraints. Specifically, reducing the constrained problem into a surrogate problem with only **linear equality constraints**, while preserving the hypergradient accuracy. In essence, we will fix the active set of constraints and linearize those constraints, thereby converting the lower-level problem into one with only equalities. By doing so, we can retain a $\mathcal{O}(\log 1/\epsilon)$ constant overhead in computing gradients. Specifically, we achieve this by first constructing the following *ghost bilevel optimization* problem:

$$\min_x \tilde{F}(x) := f(x, \tilde{y}^*) \text{ s.t. } \tilde{y}^* \in \underset{y: \tilde{h}(x,y)=0}{\arg\min} \tilde{g}(x, y), \quad \text{(P2)}$$

where we define the surrogate lower-level objective $\tilde{g}$ and combined constraints (equality constraints and linearized active inequality constraints) $\tilde{h}$ as follows:

$$\tilde{g}(x, y) = g(x, y) + \langle \lambda^*, h(x, y) \rangle + \langle \nu^*, e(x, y) \rangle, \quad \text{(3)}$$

$$\tilde{h}(x, y) = \begin{bmatrix} e(x, y) \\ \nabla_x h_{\mathcal{I}}(\bar{x}, y^*)(x - \bar{x}) + \nabla_y h_{\mathcal{I}}(\bar{x}, y^*)(y - y^*) \end{bmatrix},$$

where $\bar{x}$ denotes a given reference point (e.g., the current $x$ at which we seek the hypergradient) and $(y^*, \lambda^*, \nu^*)$ are the optimal primal-dual solution of the lower-level problem P1 at $x = \bar{x}$. Importantly, let $\mathcal{I}$ denote the *active inequality set* at the solution $y^*$, defined as

$$\mathcal{I} := \{ i \mid h_i(\bar{x}, y^*) = 0, \ \lambda_i^* > 0 \}.$$

Note that we treat the dual variable $\lambda^*(x)$ and $\nu^*(x)$ for Problem P1 as fixed constants. We add the current active inequality constraints into the objective via Lagrange terms, and simultaneously enforce those constraints by linearizing them around $x$. Thus, by this construction, Problem P2 has

only linear equality constraints in $\tilde{h}$, which enables a lower oracle complexity compared to Kornowski et al. (2024) as we proved later in Corollary 4.7.

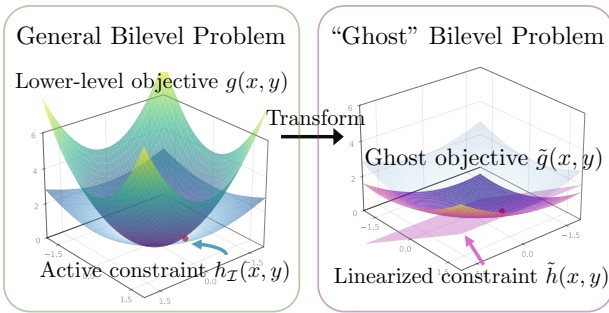

*Figure 1.* Illustration of our ghost bilevel reformulation: fix the active set $\mathcal{I}$, replace the lower-level problem with the ghost objective, and linearize the active constraints as Equation 3.

Figure 1 illustrates our method. Intuitively, Problem P2 preserves the local behavior of the original Problem P1 by focusing only on the constraints that are active at $y^*$. Inactive constraints remain slack in a neighbor of $(\bar{x}, y^*)$ and therefore do not affect the first-order behavior, whereas active constraints behave like equalities and can be captured by their first-order expansions. The method can save a lot of computation cost by avoiding dealing with all inequalities globally, but only needs to deal with several active constraints. Moreover, by our construction, at point $x = \bar{x}$, the original lower-level solution $y^*$ is feasible and also optimal for the inner minimization of Problem P2, resulting in the hypergradient at $\bar{x}$ being unchanged.

**Theorem 4.5** (Active-set equivalence). *For any given $\bar{x}$, assume that LICQ holds at a KKT point of the lower-level problem. In addition, assume that $F$ is differentiable at $\bar{x}$ and that the active set is locally constant around $\bar{x}$, Let $F$ be the upper-level objective in Problem P1 and $\tilde{F}$ be constructed as in Problem P2. Then $\nabla F(\bar{x}) = \nabla \tilde{F}(\bar{x})$.*

This theorem follows from the fact that, at $x = \bar{x}$, the ghost lower-level problem enforces the same active constraint (to first order) and uses the corresponding Lagrange objective with frozen multipliers, which yields an identical local behavior of $y^*$ and hence the same hypergradient.

Having established that the constraints can be reduced to linear equalities *without* losing hypergradient accuracy, we can now leverage the near-constant-time algorithm from Kornowski et al. (2024).

### 4.3. Finite-Difference Hypergradient Approximation

To avoid explicitly computing the $dy^*(x)/dx$ in the lower-level problem, we employ a finite-difference hypergradient strategy with an explicit approximation guarantee. The core idea is to inject a small perturbation of the upper-level ob-

jective into the lower-level problem and observe the change in the optimal solution. For a suitably chosen perturbation magnitude, this approach yields an $\epsilon$-approximate hypergradient using only first-order information.

Specifically, let $\delta > 0$ be a given small perturbation factor. We modify the lower-level objective in Problem P2 by adding a perturbation of the upper-level objective:

$$\min_x \tilde{F}(x) \coloneqq f(x, y_\delta^*)$$
$$\text{s.t. } y_\delta^* \in \underset{y:\tilde{h}(x,y)=0}{\arg\min} \; \tilde{g}(x, y) + \delta f(x, y). \quad \text{(P3)}$$

We call Problem P3 the *perturbed ghost problem* and let $(y_\delta^*, \lambda_\delta^*)$ be its primal-dual solution. The upper-level gradient can then be approximated by the following finite-difference formula derived from the perturbed Lagrangian:

$$v_x \coloneqq \frac{1}{\delta}\Big(\nabla_x[\tilde{g}(x, y_\delta^*) + \langle \lambda_\delta^*, \tilde{h}(x, y^*)\rangle] - \nabla_x[\tilde{g}(x, y^*)]\Big). \tag{4}$$

Given this finite-difference formula, we can prove that it recovers the implicit differentiation up to $\mathcal{O}(\delta)$:

$$\left\| v_x - \left(\frac{dy^*(x)}{dx}\right)^\top \nabla_y f(x, y^*(x)) \right\| \leq \mathcal{O}(\delta). \tag{5}$$

See Appendix C for proof. Intuitively, $v_x$ captures the change in the lower-level solution (and duals) with respect to $x$ through the injection of $\delta f(x, y)$, and thereby approximates $(dy^*/dx)^\top \nabla_y f(x, y^*(x))$. The full upper-level gradient is then assembled as $\tilde{\nabla} F(x) = \nabla_x f + v_x$, which is our estimate of $\nabla F(x)$. This procedure requires only first-order derivatives of $f, g, h$. Our method is summarized in Algorithm 1.

In the following, we analyze the convergence performance of our algorithm.

*Theorem* 4.6. *Under Assumptions 4.2 and 4.3, given any accuracy parameter $\epsilon > 0$, Algorithm 1 outputs $\tilde{\nabla} F$ such that $\|\tilde{\nabla} F(x) - \nabla F(x)\| \leq \epsilon$ within $\mathcal{O}(\log(1/\epsilon))$ gradient oracle evaluations.*

We compare Theorem 4.6 with the result in (Kornowski et al., 2024), specifically Theorem 5.3, which requires access to an exact dual solution and has a computation cost of $\tilde{\mathcal{O}}(\epsilon^{-1})$. By contrast, our approach only requires correct identification of the active constraints, yielding an improved complexity of $\mathcal{O}(\log(1/\epsilon))$. Combining the meta-algorithm in (Kornowski et al., 2024) with our improved rate yields the following corollary.

*Corollary* 4.7. *Under Assumptions 4.2 and 4.3, running Algorithm 1 in $\tilde{\mathcal{O}}(\delta^{-1}\epsilon^{-3})$ oracle calls converges to a $(\delta, \epsilon)$-Goldstein stationary point for the bilevel problem with linear inequality constraints.*

The rate in Corollary 4.7 matches the best known rate for non-convex non-smooth optimization in (Zhang et al., 2020a). While bilevel optimization is a special case of non-convex non-smooth optimization, naively applying their algorithms to bilevel problems requires second-order information via differentiable optimization. This underscores the significance of Corollary 4.7 as it establishes a state-of-the-art convergence rate for first-order bilevel optimization with linear inequality constraints. Furthermore, in Appendix D.1, we analyze general convex-constraint problems instead of linear constrained problems and obtain the same rate as Corollary 4.7.

---

**Algorithm 1** Inexact Gradient Oracle for Bilevel Reformulation

---

1: **Input:** Current $x$, accuracy $\epsilon$, perturbation $\delta = \mathcal{O}(\epsilon)$.
2: Compute the primal-dual pair $(y^*, \lambda^*)$ by solving $\min_{y:h(x,y)\leq 0, e(x,y)=0} g(x, y)$ with any solver
3: Compute the primal-dual pair $(y_\delta^*, \lambda_\delta^*)$ by solving $\min_{y:\tilde{h}(x,y)=0} \tilde{g}(x, y) + \delta f(x, y)$ with any solver
4: Compute the finite-difference $v_x$ as in Equation 4
5:          ▷ Approximates $(dy^*(x)/dx)^\top \nabla_y f(x, y^*)$
6: **Output:** $\tilde{\nabla} F = \nabla_x f(x, y^*) + v_x$

---

## 5. Implementation: Solver-Agnostic Differentiation

```
1   import cvxpy as cp
2   # define problem in CVXPY
3   x = cp.Variable(n)
4   u = cp.Parameter(n)
5   obj = cp.Minimize(0.5 * cp.sum_squares(x) + cp.sum(u
        * x))
6   constraints = [x == 0, x <= 1]
7   prob = cp.Problem(obj, constraints)
8   # layer = CvxpyLayer(prob, parameters=[u], variables
        =[x])
9   layer = FFOLayer(prob, parameters=[u], variables=[
        x])
10  x_star = layer(u_value)
```

*Figure 2.* Drop-in usage of our `FFOLayer` with the same interface as `CvxpyLayer`, mapping a `CVXPY` problem to a parameterized differentiable optimization layer in PyTorch.

We implement our algorithm as a PyTorch module (Paszke et al., 2017) that can be seamlessly integrated into existing codebases, as shown in Figure 2. In particular, it is compatible with existing differentiable optimization layers (e.g., `CvxpyLayer`), where the embedded solver is a *black box* for the lower-level problem: given $x$ and problem functions $(g, h, e)$, it returns $y^*(x)$, after which the layer evaluates $f(x, y^*(x))$ and backpropagates through $f$. This design keeps the upper-level objective $f$ separate from the solver.

However, Line 3 in Algorithm 1 requires solving the perturbed lower-level problem with the objective $\tilde{g}(x, y) +$

$\delta f(x, y)$, which explicitly depends on $f$. Under the black-box solver interface, the solver is not provided the explicit form of $f$, so this perturbed solve is not a direct drop-in and requires restructuring the lower-level formulation.

**Solver-agnostic reformulation** We resolve this issue by constructing the following surrogate upper-level objective:

$$\hat{f}(x, y) := c^\top y, \ \hat{F}(x) := \hat{f}(x, y^*) = c^\top y^*, \quad (6)$$

where $c := \texttt{detach}(\nabla_y f(x, y^*(x)))$[1]. Since $\nabla_y f = c = \nabla_y \hat{f}$, this replacement preserves the hypergradient we seek. See Appendix E for proof.

In addition, with this transformation, the perturbed lower-level problem of Problem P3 can be implemented by simply adding a linear term:

$$\hat{y}_\delta^* \in \underset{y:\tilde{h}(x,y)=0}{\arg\min} \ \tilde{g}(x, y) + \delta c^\top y. \quad (7)$$

Concretely, we first solve the original lower-level problem to obtain $y^*$ (Algorithm 1, Line 2). We then compute $c = \nabla_y f(x, y^*(x))$ via *automatic differentiation* and stop gradients through $c$. Finally, we solve the perturbed lower-level problem 7, to obtain $\hat{y}_\delta^*$, which is used in the finite-difference approximation (Equation 4). This procedure effectively incorporates the required upper-level information into the lower-level perturbation while avoiding any objective-specific derivations.

Therefore, this makes our method *solver-agnostic*: we treat the lower-level solver as a black box, and changing the solver does not require implementing a new, solver-specific backward pass. In the backward step, we only need to solve the perturbed problem using the incoming gradient signal $c$, and then perform the finite difference. Thus, the gradient computation remains unchanged across solvers, which enables us to use many advanced convex solvers such as GUROBI and MOSEK.

In our code implementation, we provide two variants of `FFOLayer` (while our theory covers both cases uniformly): `FFOCP`: our method applied to general convex problems; `FFOQP`: a specialization of our method for QP layers, which exploits the quadratic structure (e.g., having a closed-form solution for QP). Both variants use the same underlying algorithm, but `FFOQP` is optimized for the QP case.

# 6. Experiments

We evaluate the performance of our fully first-order bilevel method on a variety of tasks, comparing against several baseline methods for differentiable optimization layers. The baselines include:

- `CvxpyLayer` (Agrawal et al., 2019a): a general differentiable convex optimization layer specified in CVXPY; problems are compiled to cone programs and solved by `diffcp` (Agrawal et al., 2019b).
- `QPTH` (Amos & Kolter, 2017): a differentiable QP layer based on a primal-dual Newton (interior-point) method.
- `LPGD` (Paulus et al., 2024): a first-order primal-dual solver that applies proximal gradient steps to the conic form of the problem. It estimates gradients by finite difference of Lagrange proximal solutions, which is also a first-order method.
- `BPQP` (Pan et al., 2024): a general differentiable convex layer that transforms a convex program into a QP, which is then solved by efficient QP solvers such as OSQP (Stellato et al., 2020).
- `dQP` (Magoon et al., 2025): a recent black-box differentiable layer for QPs that computes hypergradients using a reduced KKT system based on the active set.

For all baselines, we use official releases of `CvxpyLayer`, `QPTH`, `LPGD`, and `dQP` (with GUROBI backend), and implement our own version of `BPQP`. For our methods, we use SCS (O'Donoghue et al., 2016) as the forward and backward solver for `FFOCP`, and qpsolver (Caron et al., 2025) as the forward solver for `FFOQP`. Due to the lack of GPU solvers that support the batch settings, we run all methods on CPU. Detailed experiment settings can be found in Appendix G.

## 6.1. Synthetic Decision-Focused Task

Our first task is a synthetic decision-focused learning task (Mandi et al., 2024), which naturally fits a bilevel formulation. We generate a synthetic dataset of $N$ samples $\{x_i, y_i\}_{i=1}^N$, where each $y_i \in \mathbb{R}^{d_y}$ represents a ground-truth decision and $x_i \in \mathbb{R}^{d_x}$ is an input feature vector. We then train a model $q_\theta(x_i)$ to first predict the linear coefficients of a lower-level quadratic program, and the solution to the quadratic program $y_\theta^*(x_i)$ is fed into a linear loss function $y_i^\top y_\theta^*(x_i)$. The task is to learn the neural network parameter $\theta$ to minimize $\sum_{i=1}^N y_i^\top y_\theta^*(x_i)$. The synthetic DFL problem is formulated as follows:

$$\min_\theta \sum_{i=1}^N y_i^\top y_\theta^*(x_i)$$

$$\text{s.t. } y_\theta^*(x_i) \in \arg\min_{y:Gy\leq h} \frac{1}{2} y^\top Q y - q_\theta(x_i)^\top y,$$

where $q_\theta(x_i)$ is the predicted linear coefficients for the $i$-th QP, and the QP has fixed matrix $Q$ and constraints $Gy \leq h$.

## 6.2. Sudoku Task

Our second task is adopted from the Sudoku task proposed by (Amos & Kolter, 2017). Here, solving a Sudoku puzzle can be approximately represented as solving a linear program. Given a dataset of partially filled $n \times n$ Sudoku

---

[1] `detach(·)` denotes the stop-gradient operator.

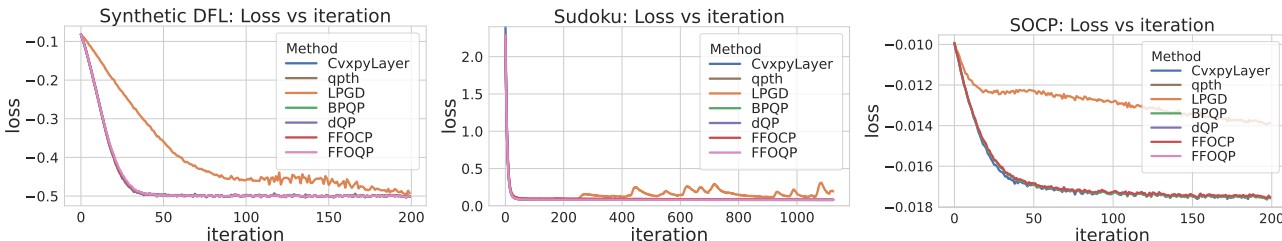

*Figure 3.* Training convergence for the synthetic DFL task, Sudoku, and SOCP task. Our methods (FFOCP and FFOQP) have the same convergence performance as other exact gradient methods.

puzzles $p_i \in \{0,1\}^{n^3}$ and their solutions $y_i \in \{0,1\}^{n^3}$, the task is to learn the rules of Sudoku puzzles, which are the linear constraint parameters $A(\theta)$ and $b(\theta)$ of the linear program. The Sudoku problem is formulated as follows:

$$\min_\theta \sum_{i=1}^N \|y_i - y^*(\theta, p_i)\|_2^2$$

$$\text{s.t. } y^*(\theta, p_i) \in \arg\min_{y: A(\theta)y=b(\theta), y \geq 0} \frac{\tau}{2} y^\top y - p_i^\top y.$$

Essentially, the solver must learn the Sudoku rules (constraints) so that the solution completes any puzzle $p_i$. We set $n^3 = 729$ and $\tau = 0.1$ as a small quadratic perturbation.

### 6.3. Second-order Cone Programming (SOCP)

In order to test our proposed method in *general convex constraints*, we include a second-order cone program, which can be regarded as robust programming. This task is formulated as a decision-focused problem similar to the QP case, but with the added complexity of a second-order ($l_2$ norm) constraint. Formally, the lower-level problem is an SOCP:

$$\min_{y: Gy \leq h, \|y\|_2 \leq c} \frac{1}{2} y^\top Q y - q_\theta(x_i)^\top y.$$

This SOCP task allows us to assess how our method and baselines perform on an optimization layer with nonlinear constraints and whether our first-order approach retains its advantages in this setting.

## 7. Results and Discussion

**Convergence behavior** We first evaluate training loss convergence. Figure 3 shows the results on the three benchmark tasks: the synthetic DFL, Sudoku, and SOCP tasks. Across all tasks, we observe that both of our proposed first-order methods closely match the optimization behavior of exact solvers (CvxpyLayer and QPTH) in terms of training loss reduction. This suggests that replacing implicit differentiation with our first-order approximate hypergradient oracle does not degrade optimization performance on these tasks.

Note that we found that LPGD's reported tolerance ($\epsilon = 10^{-4}$) in their paper was insufficient for convergence on the synthetic and Sudoku tasks. We demonstrate this finding

in Appendix G.2. Thus, in Figure 3, we report LPGD's performance with the same tolerance as ours, which indicates that our solvers consistently outperform the prior first-order solver. This advantage comes from the explicit non-asymptotic guarantee for the computed hypergradient, which makes the backward pass more stable and less sensitive to the solver tolerance in practice.

**Solving time comparison with KKT-based methods** We next investigate the runtime of our approach versus the baselines. Figure 4 plots the averaged forward (layer solving) and backward (gradient computation) times for each method. We find that our first-order approach has significant runtime advantages in scenarios where the KKT-based methods struggle with numerical stability.

For example, CvxpyLayer's backward pass is extremely slow and unstable in the Sudoku task (which yields a dense, ill-conditioned KKT matrix), whereas FFOCP's backward pass remains fast and robust. In general, when the problem is ill-conditioned or very large-scale, factorizing or inverting the Hessian becomes computationally expensive for KKT-based methods. On the other hand, our method avoids explicit KKT formulation and obtains hypergradients via only two optimization solvers followed by a finite-difference approximation, which yields better robustness to ill-conditioning.

For the forward pass, FFOCP is also faster than CvxpyLayer because CvxpyLayer incurs a heavy canonicalization overhead while solving the problem.

In QP tasks, FFOCP achieves comparable or even better performance than QPTH despite not leveraging the QP structure. QPTH achieves a faster forward time in Sudoku and has a much lower backward time in all QP tasks since factorization of the KKT matrix is done in the forward pass. However, QPTH has a much higher forward time in synthetic DFL. When we specialize our algorithm to QP, FFOQP is superior to all KKT-based methods in synthetic QP and has a runtime similar to QPTH in Sudoku.

**Solving time comparison with optimization-based methods** We also compare our approach to recent first-order methods (LPGD) and black-box differentiation methods (BPQP, dQP). These methods compute the hypergradient by iteratively solving optimization problems, which can

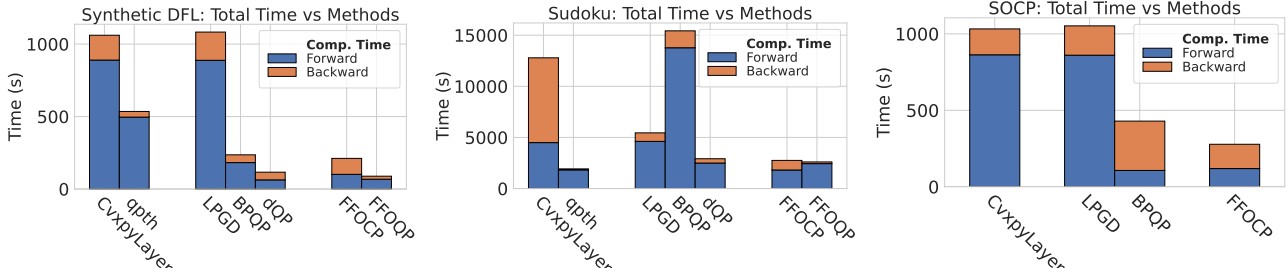

*Figure 4.* Computation costs for the synthetic DFL task, Sudoku, and SOCP task for variable dimension $d_y = 800$. Our convex solver FFOCP outperforms other convex solvers: CvxpyLayer, LPGD, and BPQP, while our QP solver FFOQP outperforms other QP solvers.

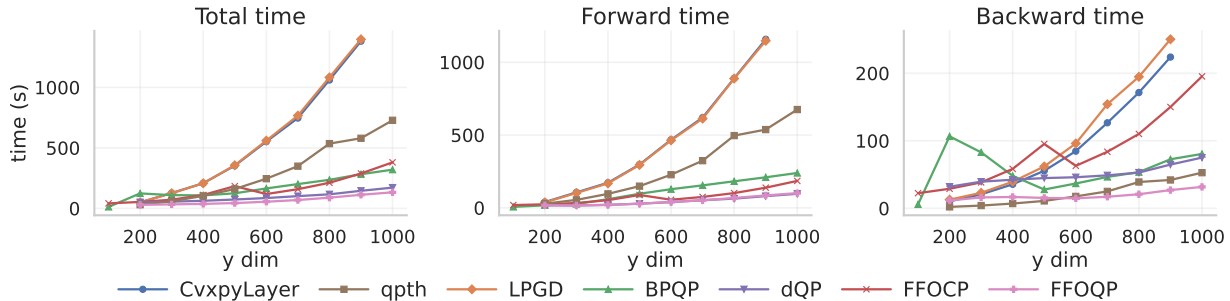

*Figure 5.* Different variable dimension for synthetic DFL. FFOCP and FFOQP exhibit sublinear computation time scaling over the problem dimension. (It does not include the results for CvxpyLayer and LPGD at $d_y = 1000$ due to out of memory.)

introduce significant overhead.

For LPGD, its forward passes are much slower than ours because it is built on diffcp, which requires additional canonicalization and can be slow in large-scale cases. The other two baselines, BPQP and dQP, both avoid explicit differentiation by transforming the original problem into a QP, which allows using efficient black-box QP solvers. However, this transformation significantly increases the problem dimension and complexity of each backward pass. For example, BPQP must compute certain second-order information to construct the equivalent QP, which becomes the main bottleneck for large problems. In contrast, FFOCP works on the problem in its original form and uses only first-order information. As shown in Figure 4, FFOCP is faster than BPQP and dQP, despite their use of specialized QP solvers. When using the QP-specialized variant, FFOQP, our method outperforms all optimization-based methods on QP tasks.

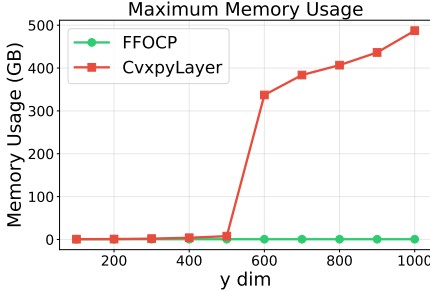

*Figure 6.* Maximum memory usage during training with different variable dimensions. FFOCP exhibits near-constant peak memory, while CvxpyLayer's memory cost grows dramatically with $d_y$.

**Scalability with problem size** A key motivation for our approach is better scaling to large problem sizes. To test this, we conduct two ablation studies by varying the dimension of the decision variable $y$ in both QP and SOCP tasks. Figure 5 reports the average total, forward, and backward time as $d_y$ grows. The ablation confirms that our methods exhibit much slower growth in computation time with increasing dimension compared to baseline methods. Importantly, Figure 6 shows that our solver remains memory-light across all variable dimensions, while CvxpyLayer's memory increases rapidly and becomes prohibitive at large $d_y$ (eventually leading to out-of-memory at $d_y = 1000$).

These trends align with our complexity analysis: by eliminating Hessian inversions, FFOLayer backpropagation cost grows sublinear in problem size, as opposed to superlinear time and memory growth for baselines.

**Other ablation studies** We also discuss more ablation results on QPTH with GPU, the distance and cosine similarity between the approximated gradient and exact gradient, solver tolerances, and batch sizes in Appendix G.2.

## 8. Conclusion

We introduce a fully first-order differentiation method for differentiable optimization layers. Our approach reformulates the differentiable optimization as a bilevel problem and introduces an active-set Lagrangian oracle to compute hypergradients using only gradients and function evaluations. Theoretically, we show that an approximate hypergradient can be obtained in $\mathcal{O}(\log(1/\epsilon))$ time per iteration, leading to an overall convergence complexity of $\mathcal{O}(\delta^{-1}\epsilon^{-3})$

for constrained bilevel optimization. Experimental results demonstrate comparable convergence to exact layers with substantially improved scalability in well-clock time and peak memory as problem size increases.

## Acknowledgements

This project was supported by the Schmidt Sciences AI2050 Fellowship, NSF grant IIS-2403240, and NIH grant R01HL184139.

## Impact Statement

This paper proposes `FFOLayer`, a fully first-order method for differentiating convex optimization layers using only gradients and function evaluations. It can reduce the cost of using embedded convex optimizations in ML pipelines (e.g., structured prediction, control, and meta-learning). Potential risks come mainly from how the layer is used in downstream applications (e.g., poorly specified objectives and constraints), so we recommend domain-specific validation when deployed in high-stakes settings.

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

## Appendix

## A. Lemmas for Linear Equality Constraints

In this section, we consider a bilevel problem with only linear equality constraints:

$$F(x) := f(x, y^*) \quad \text{s.t.} \quad y^* \in \underset{y: Ax - By - b = 0}{\arg\min} g(x, y), \tag{8}$$

We derive several useful lemmas here, which will then be used in the proofs for the general convex constraints. The first lemma offers an intuitive interpretation of the gradient as composed of two components: one behaves as if the problem is unconstrained but projected onto the kernel of the active constraints; the other arises solely from the constraints, projected onto the span of the active constraints. We will make extensive use of this form in the subsequent analysis.

*Lemma* A.1. *Consider the lower-level problem with linear equality constraints with a full row rank* B, *the following holds:*

$$\frac{dy^*}{dx} = \Pi_B^G\big(-[\nabla_{yy}g(x, y^*)]^{-1}\nabla_{yx}g(x, y^*)\big) + (I - \Pi_B^G)(B^\dagger A) . \tag{9}$$

*where we define $G = \nabla_{yy}g(x, y^*)$ and $\Pi_B^G(z) := \arg\min_{By=0}(y - z)^\top G(y - z)$ as the projection operator onto the null space of* B.

*Proof.* Denote by $\frac{dy^*}{dx} = d_y$ and $\frac{d\lambda^*}{dx} = d_\lambda$. The KKT system can be written as,

$$Gd_y - B^\top d_\lambda = -\nabla_{yx}g(x, y^*) \ ; \ Bd_y = A .$$

This coincides with the KKT conditions of the following problem,

$$\min_{d_y} \frac{1}{2}d_y^\top Gd_y + \nabla_{yx}g(x, y^*)d_y \quad \text{s.t.} \quad Bd_y = A .$$

Denote by $\langle a, b\rangle_G = a^\top Gb$ and $\|a\|_G = \sqrt{\langle a, a\rangle_G}$. The above problem can be rewritten as,

$$\min_{d_y} \|d_y + G^{-1}\nabla_{yx}g(x, y^*)\|_G^2 \quad \text{s.t.} \quad \langle G^{-1}B^\top, d_y\rangle_G = A .$$

We can decompose each vector $v$ into $v^\|$ and $v^\perp$ such that $v^\|$ and $v^\perp$ lies in the span and null space of $G^{-1}B^\top$ respectively. The problem then simplifies to,

$$\min_{d_y^\|, d_y^\perp} \|d_y^\| + \big(G^{-1}\nabla_{yx}g(x, y^*)\big)^\|\|_G^2 + \|d_y^\perp + \big(G^{-1}\nabla_{yx}g(x, y^*)\big)^\perp\|_G^2 \quad \text{s.t.} \quad \langle G^{-1}B^\top, d_y^\|\rangle_G = A \ ; \ d_y^\| \in \text{span}(G^{-1}B^\top) .$$

Note that $d_y^\perp$ is unconstrained, and thus $d_y^\perp = -\big(G^{-1}\nabla_{yx}g(x, y^*)\big)^\perp = \Pi_B^G\big(-[\nabla_{yy}g(x, y^*)]^{-1}\nabla_{yx}g(x, y^*)\big)$. In addition, the feasible set for $d_y^\|$ consists of a unique point. The solution can be constructed by taking any point on $\langle G^{-1}B^\top, d_y^\|\rangle_G = A$ and projecting onto the span of $G^{-1}B^\top$. One such point is $B^\dagger A$ and, therefore, $d_y^\| = \big(B^\dagger A\big)^\| = (I - \Pi_B^G)(B^\dagger A)$. This concludes the proof. □

*Lemma* A.2. *Given $\mu_g I \preceq G \preceq C_g I$ and let $\kappa_g = C_g/\mu_g$, we have $\|\Pi_B^G\| \leq 1 + \sqrt{\kappa_g}$.*

*Proof.* For a given $x$, let $y = \Pi_B^G(x)$. Since $B0 = 0$, we have,

$$\mu_g\|x - y\|^2 \leq (x - y)^\top G(x - y) \leq x^\top Gx \leq C_g\|x\|^2 .$$

Therefore, $\|x - y\| \leq \sqrt{\kappa_g}\|x\|$. By the triangle inequality, $\|y\| \leq \|x\| + \|x - y\| \leq (1 + \sqrt{\kappa_g})\|x\|$. □

*Lemma* A.3. *Given $\mu_g I \preceq G, \tilde{G} \preceq C_g I$ and $\kappa_g = C_g/\mu_g$, we have $\|\Pi_B^G - \Pi_B^{\tilde{G}}\| \leq \sqrt{\frac{\|G - \tilde{G}\|}{\mu_g}}\kappa_g$.*

*Proof.* For a given $x$, let $y = \Pi_B^G(x)$ and $\tilde{y} = \Pi_B^{\tilde{G}}(x)$. Since the function $(y-x)G(y-x)$ is $\mu_g$-strongly convex in $y$, we have

$$\mu_g\|y - \tilde{y}\|^2 + (y-x)G(y-x) \leq (\tilde{y}-x)G(\tilde{y}-x) .$$

Similarly, we have

$$\mu_g\|y - \tilde{y}\|^2 + (\tilde{y}-x)\tilde{G}(\tilde{y}-x) \leq (y-x)\tilde{G}(y-x) .$$

Summing up the two inequalities we get,

$$
\begin{aligned}
2\mu_g\|y - \tilde{y}\|^2 &\leq (y-x)(\tilde{G}-G)(y-x) + (\tilde{y}-x)(G-\tilde{G})(\tilde{y}-x) \\
&\leq \|G-\tilde{G}\|\big[\|y-x\|^2 + \|\tilde{y}-x\|^2\big] \\
&\leq \|G-\tilde{G}\| \cdot 2\kappa_g\|x\|^2
\end{aligned}
$$

Taking the square root on both sides yields the desired bound. $\qquad\square$

*Lemma* A.4. *The solution $y^*$ is $L_y$-Lipschitz continuous and $C_y$-smooth as a function of $x$, where $L_y = \mathcal{O}(\sqrt{\kappa_g} \cdot (\kappa_g + \|B^\dagger A\|))$ and $C_y = \mathcal{O}(\sqrt{\frac{S_g}{\mu_g}} \cdot \kappa_g(\kappa_g\sqrt{\frac{S_g}{\mu_g}} + \kappa_g + \|B^\dagger A\|)^2)$. Thus, the hyper-objective is gradient-Lipschitz with a smoothness constant of $C_F := \mathcal{O}(C_f L_y^2 + L_f C_y)$.*

*Proof.* From Lemma A.1 and Lemma A.2, we immediately get

$$\left\|\frac{dy^*}{dx}\right\| \leq (1 + \sqrt{\kappa_g}) \cdot \left(\frac{C_g}{\mu_g} + \|B^\dagger A\|\right) .$$

For a given pair $(x, \bar{x})$, let us denote the corresponding optimal solutions by $y^*$ and $\tilde{y}$. In addition, let $G = \nabla_{yy}g(x, y^*)$ and $\tilde{G} = \nabla_{yy}g(\bar{x}, \tilde{y})$. Then, we have

$$
\begin{aligned}
&\left\|\frac{dy^*(x)}{dx} - \frac{dy^*(\bar{x})}{dx}\right\| \\
&\leq \left\|\Pi_B^G\big(G^{-1}\nabla_{yx}g(x, y^*) - \tilde{G}^{-1}\nabla_{yx}g(\bar{x}, \tilde{y})\big)\right\| && (10) \\
&\quad + \left\|\big(\Pi_B^G - \Pi_B^{\tilde{G}}\big)\tilde{G}^{-1}\nabla_{yx}g(\bar{x}, \tilde{y})\big)\right\| && (11) \\
&\quad + \left\|\big(\Pi_B^G - \Pi_B^{\tilde{G}}\big)B^\dagger A\right\| && (12)
\end{aligned}
$$

Equation (10) can be bounded by $(1 + \sqrt{\kappa_g}) \cdot \left(\frac{S_g}{\mu_g}(1 + L_y) + \frac{C_g S_g}{\mu_g^2}(1 + L_y)\right)$.

Since $\|G - \tilde{G}\| \leq S_g(1 + L_y)$, Equation (11) and Equation (12) can be bounded by $\sqrt{\frac{S_g(1+L_y)}{\mu_g}} \cdot \kappa_g \cdot (\kappa_g + \|B^\dagger A\|)$. Summing up all terms and with some simplification, we obtained the desired result. $\qquad\square$

# B. Proofs and Technical Details for Section 4.1

*Theorem* B.1. *For any given $\bar{x}$, let $F$ be the upper-level objective in Problem P1 and $\tilde{F}$ be constructed as in Problem P2. Assume LICQ holds at a KKT point of the lower-level problem. If $F$ is differentiable at $\bar{x}$ and the active set is locally constant around $\bar{x}$, then $\nabla F(\bar{x}) = \nabla \tilde{F}(\bar{x})$.*

*Proof.* It is straightforward to check that the point $y^*$ satisfies the KKT stationarity conditions:

$$
\begin{aligned}
\nabla_y \tilde{g}(\bar{x}, y^*) + 0^\top \nabla_y \tilde{h}(\bar{x}, y^*) &= 0, \\
\tilde{h}(\bar{x}, y^*) &= 0. && (13)
\end{aligned}
$$

Therefore, $y^*$ is also optimal for the new inner problem, i.e., $\tilde{y}^* = y^*$. Intuitively, the surrogate inner problem has the same local optimality conditions as the true inner problem at the active constraints, so it yields the same total derivative $dy^*/dx$.

Formally, under the LICQ assumption, $\tilde{\mathsf{B}} = \nabla_y \tilde{h}(\bar{x}, y^*) = \begin{bmatrix} \nabla_y e(\bar{x}, y^*) \\ \nabla_y h_{\mathcal{I}}(\bar{x}, y^*) \end{bmatrix}$ has full row rank. The KKT system for the ghost problem is

$$\begin{pmatrix} \nabla_{yy}\tilde{g}(\bar{x}, y^*) & \tilde{\mathsf{B}}^\top \\ \tilde{\mathsf{B}} & 0 \end{pmatrix} \begin{pmatrix} \frac{d\tilde{y}^*(\bar{x})}{dx} \\ \frac{d(\tilde{\nu}^*(\bar{x}), \tilde{\lambda}_{\mathcal{I}}^*(\bar{x}))}{dx} \end{pmatrix} = \begin{pmatrix} -\nabla_{yx}\tilde{g}(\bar{x}, y^*) \\ -\nabla_x \begin{bmatrix} e(\bar{x}, y^*) \\ h_{\mathcal{I}}(\bar{x}, y^*) \end{bmatrix} \end{pmatrix}. \tag{14}$$

Compare this with the KKT system for Problem P1:

$$\begin{pmatrix} \nabla_{yy}\tilde{g}(\bar{x}, y^*) & \tilde{\mathsf{B}}^\top \\ \mathrm{diag}(\mathbf{1}, \lambda_{\mathcal{I}}^*)\tilde{\mathsf{B}} & 0 \end{pmatrix} \begin{pmatrix} \frac{dy^*(\bar{x})}{dx} \\ \frac{d(\nu^*(\bar{x}), \lambda_{\mathcal{I}}^*(\bar{x}))}{dx} \end{pmatrix} = \begin{pmatrix} -\nabla_{yx}\tilde{g}(\bar{x}, y^*) \\ -\mathrm{diag}(\mathbf{1}, \lambda_{\mathcal{I}}^*)\nabla_x \begin{bmatrix} e(\bar{x}, y^*) \\ h_{\mathcal{I}}(\bar{x}, y^*) \end{bmatrix} \end{pmatrix}. \tag{15}$$

By simple linear algebra and using the fact that $\mathrm{diag}(\lambda_{\mathcal{I}}^*) \succ 0$, we obtain $\frac{d\tilde{y}^*(\bar{x})}{dx} = \frac{dy^*(\bar{x})}{dx}$ and therefore $\nabla F(\bar{x}) = \nabla \tilde{F}(\bar{x})$. $\qquad\square$

# C. Proofs and Technical Details for Section 4.3

*Lemma* C.1 (Finite-difference approximation error). *Fix point $x$ and let $y^*(x)$ denote the primal solution of the* original *lower-level problem (Problem P1) at this $x$. Assume $\tilde{g}, \tilde{h}, f$ are twice continuously differentiable in $(x, y)$. Let $\tilde{y}^*(x)$ solve the ghost lower-level problem (Problem P2) $\tilde{y}^*(x) \in \arg\min_{y: \tilde{h}(x,y)=0} \tilde{g}(x, y)$, and let $\tilde{\lambda}(x)$ denote its associated dual variable. For $\delta > 0$, let $(\tilde{y}_\delta^*(x), \tilde{\lambda}_\delta^*(x))$ satisfy the KKT conditions of the perturbed ghost problem (Problem P3). Assume the KKT system at $\delta = 0$ is strongly regular, so that $\delta \mapsto (\tilde{y}_\delta^*(x), \tilde{\lambda}_\delta^*(x))$ is differentiable at $\delta = 0$. Let the finite-difference estimator defined as in Equation 4 :*

$$v_x := \frac{1}{\delta}\Big( \nabla_x[\tilde{g}(x, y_\delta^*) + \langle \lambda_\delta^*, \tilde{h}(x, y^*) \rangle] - \nabla_x[\tilde{g}(x, y^*)] \Big). \tag{16}$$

*Then, under Assumptions 4.2 and 4.3, there exists $C > 0$ such that for all sufficiently small $\delta$,*

$$\left\| v_x - \left( \frac{dy^*(x)}{dx} \right)^\top \nabla_y f(x, y^*(x)) \right\| \le C\,\delta. \tag{17}$$

*Proof.* Since $\nabla_x \langle \lambda, \tilde{h}(x, \tilde{y}^*) \rangle = \nabla_x \tilde{h}(x, \tilde{y}^*)^\top \lambda$, we rewrite

$$v_x(\delta) = \frac{1}{\delta}\Big( \nabla_x \tilde{g}(x, \tilde{y}_\delta^*) - \nabla_x \tilde{g}(x, \tilde{y}^*) \Big) + \nabla_x \tilde{h}(x, \tilde{y}^*)^\top \frac{\tilde{\lambda}_\delta^* - \tilde{\lambda}}{\delta}. \tag{18}$$

Using the Taylor expansion of $\nabla_x \tilde{g}(x, \cdot)$ around $\tilde{y}^*$ and get

$$\tilde{y}_\delta^*(x) = \tilde{y}^*(x) + \delta \frac{d\tilde{y}_\delta^*(x)}{d\delta}\Big|_{\delta=0} + O(\delta^2), \qquad \tilde{\lambda}_\delta^*(x) = \tilde{\lambda}(x) + \delta \frac{d\tilde{\lambda}_\delta^*(x)}{d\delta}\Big|_{\delta=0} + O(\delta^2). \tag{19}$$

Therefore,

$$\tilde{y}_\delta^* - \tilde{y}^* = \delta \frac{d\tilde{y}_\delta^*(x)}{d\delta}\Big|_{\delta=0} + O(\delta^2), \tag{20}$$

and we can obtain

$$\frac{1}{\delta}\Big( \nabla_x \tilde{g}(x, \tilde{y}_\delta^*) - \nabla_x \tilde{g}(x, \tilde{y}^*) \Big) = \nabla_{xy}^2 \tilde{g}(x, \tilde{y}^*) \frac{d\tilde{y}_\delta^*(x)}{d\delta}\Big|_{\delta=0} + O(\delta). \tag{21}$$

Similarly,

$$\tilde{\lambda}_\delta^* - \tilde{\lambda} = \delta \frac{d\tilde{\lambda}_\delta^*(x)}{d\delta}\Big|_{\delta=0} + O(\delta^2) \tag{22}$$

implies

$$\nabla_x \tilde{h}(x, \tilde{y}^*)^\top \frac{\tilde{\lambda}_\delta^* - \tilde{\lambda}}{\delta} = \nabla_x \tilde{h}(x, \tilde{y}^*)^\top \frac{d\tilde{\lambda}_\delta^*(x)}{d\delta}\Big|_{\delta=0} + O(\delta). \tag{23}$$

Hence

$$v_x(\delta) = \nabla_{xy}^2 \tilde{g}(x, \tilde{y}^*) \frac{d\tilde{y}_\delta^*(x)}{d\delta}\Big|_{\delta=0} + \nabla_x \tilde{h}(x, \tilde{y}^*)^\top \frac{d\tilde{\lambda}_\delta^*(x)}{d\delta}\Big|_{\delta=0} + O(\delta). \tag{24}$$

Next, differentiating the perturbed ghost KKT system with respect to $\delta$ at $\delta = 0$ yields the linearized KKT system

$$\underbrace{\begin{bmatrix} \nabla_{yy}^2 \tilde{g}(x, \tilde{y}^*) & \nabla_y \tilde{h}(x, \tilde{y}^*)^\top \\ \nabla_y \tilde{h}(x, \tilde{y}^*) & 0 \end{bmatrix}}_{=:K(x)} \begin{bmatrix} \frac{d\tilde{y}_\delta^*(x)}{d\delta}\big|_{\delta=0} \\ \frac{d\tilde{\lambda}_\delta^*(x)}{d\delta}\big|_{\delta=0} \end{bmatrix} = - \begin{bmatrix} \nabla_y f(x, \tilde{y}^*) \\ 0 \end{bmatrix}, \tag{25}$$

where we used $\tilde{\lambda}(x) = 0$. On the other hand, differentiating the unperturbed ghost KKT system with respect to $x$ gives

$$K(x) \begin{bmatrix} \frac{d\tilde{y}^*(x)}{dx} \\ \frac{d\tilde{\lambda}(x)}{dx} \end{bmatrix} = - \begin{bmatrix} \nabla_{xy}^2 \tilde{g}(x, \tilde{y}^*) \\ \nabla_x \tilde{h}(x, \tilde{y}^*) \end{bmatrix}. \tag{26}$$

Since $K(x)$ is symmetric, we have the adjoint identity

$$\left( \frac{d\tilde{y}^*(x)}{dx} \right)^\top \nabla_y f(x, \tilde{y}^*) = - \left[ \nabla_{xy}^2 \tilde{g}(x, \tilde{y}^*) \ \nabla_x \tilde{h}(x, \tilde{y}^*)^\top \right] K(x)^{-1} \begin{bmatrix} \nabla_y f(x, \tilde{y}^*) \\ 0 \end{bmatrix}. \tag{27}$$

Combining Equation (25) with Equation (24) gives

$$v_x(\delta) = - \left[ \nabla_{xy}^2 \tilde{g}(x, \tilde{y}^*) \ \nabla_x \tilde{h}(x, \tilde{y}^*)^\top \right] K(x)^{-1} \begin{bmatrix} \nabla_y f(x, \tilde{y}^*) \\ 0 \end{bmatrix} + O(\delta). \tag{28}$$

Comparing with Equation (27) yields

$$\left\| v_x(\delta) - \left( \frac{d\tilde{y}^*(x)}{dx} \right)^\top \nabla_y f(x, \tilde{y}^*(x)) \right\| \leq C\,\delta. \tag{29}$$

Finally, using the identities from Theorem B.1 that $\tilde{y}^*(x) = y^*(x)$ and $\frac{d\tilde{y}^*(x)}{dx} = \frac{dy^*(x)}{dx}$ at the point $x$, we obtain Equation (17). $\qquad \square$

**Theorem** C.2. *Under Assumptions 4.2 and 4.3, given any accuracy parameter $\epsilon > 0$, Algorithm 1 outputs $\tilde{\nabla}F$ such that $\|\tilde{\nabla}F(x) - \nabla F(x)\| \leq \epsilon$ within $\tilde{\mathcal{O}}(1)$ gradient oracle evaluations*

*Proof.* Let $\tilde{\delta} = \mathcal{O}(\epsilon^2)$. Since both Problem P0 and Problem P3 are well-conditioned, we can obtain $\tilde{\delta}$-close primal solutions $(\tilde{y}, \tilde{y}_\delta)$ in $\tilde{\mathcal{O}}(1)$ time. Under the assumption of active-set identification, the reduction is exact and $\tilde{y}$ is also a $\tilde{\delta}$-close solution to the inner problem of Problem P2. By Lemma A.7 in (Kornowski et al., 2024), $(\tilde{\lambda}, \tilde{\lambda}_\delta) = ((\tilde{B}^\top)^\dagger \nabla_y \tilde{g}(x, \tilde{y}), (\tilde{B}^\top)^\dagger \nabla_y \tilde{g}(x, \tilde{y}_\delta))$ are $\mathcal{O}(\tilde{\delta})$-close dual solutions.

Next, we follow the proof of Theorem 3.1 in (Kornowski et al., 2024). Let $\tilde{v}_x$ be constructed as in Equation 4, but with exact solutions replaced by the corresponding approximate ones. Define $\mathsf{A} = \begin{bmatrix} \nabla_x e(x, y) \\ \nabla_x h(x, y) \end{bmatrix}$ and $\tilde{\mathsf{A}} = \begin{bmatrix} \nabla_x e(x, y) \\ \nabla_x h_\mathcal{I}(x, y) \end{bmatrix}$. We have

$$\|v_x - \tilde{v}_x\| \leq \frac{1}{\delta} \Big( \|\nabla_x \tilde{g}(x, y_\delta^*) - \nabla_x \tilde{g}(x, \tilde{y}_\delta)\| + \|\nabla_x \tilde{g}(x, \tilde{y}^*) - \nabla_x \tilde{g}(x, y^*)\| + \|\tilde{A}^\top (\lambda_\delta^* - \tilde{\lambda}_\delta)\| + \|\tilde{A}^\top (\tilde{\lambda} - \lambda^*)\| \Big)$$

$$\leq \frac{2}{\delta} [C_g + \|A\|]\, \tilde{\delta} \ \leq \ O(\delta).$$

From Lemma 3.2 in (Kornowski et al., 2024), we also have $\left\| v_x - \left( \frac{dy^*(x)}{dx} \right)^\top \nabla_y f(x, y^*) \right\| \leq \mathcal{O}(\delta)$. Therefore,

$$\|\tilde{\nabla}F(x) - \nabla F(x)\| \leq \|\nabla_x f(x, \tilde{y}) - \nabla_x f(x, y^*)\| + \|v_x - \tilde{v}_x\| + \left\| v_x - \left( \frac{dy^*(x)}{dx} \right)^\top \nabla_y f(x, y^*) \right\| \ \leq O(\delta).$$

$\qquad \square$

# D. Proofs and Technical Details for General Convex Constraints

## D.1. Hardness for General Convex Constraints

In the previous discussion, we studied the linear inequality constraints. Here, we are interested in general convex constraints, where the upper-level landscape can be drastically more complicated than the lower-level problem suggests. The following example demonstrates that even with a single quadratic (convex) constraint, the resulting value function may fail to be smooth or Lipschitz near the optimum. Consider the bilevel problem with convex constraints:

$$f(x, y) = y, \quad g(x, y) = (y - 2)^2, \quad h(x, y) = x^2 + y^2 - 1. \tag{30}$$

Although the lower-level problem has only a simple quadratic constraint, the overall problem is equivalent to minimizing $F(x) = \sqrt{1 - x^2}$, which is neither convex nor smooth and is not even Lipschitz near the optimum $x^* = 1$. In other words, while the bilevel formulation has seemingly simple convex constraints, the overall optimization is nontrivial and has an unbounded gradient norm. This example motivates us to introduce additional assumptions to ensure bounded solutions and hypergradients in the general convex constraint setting.

*Assumption* D.1. We additionally assume

1. We have access to an approximate oracle that returns a solution $(\tilde{y}, \tilde{\lambda})$ such that $\|\tilde{y} - y^*\| + \|\tilde{\lambda} - \lambda^*\| \le \epsilon$, and shares the same active constraints as the optimal one.

2. The constraint $h$ is $L_h$-Lipschitz continuous, $C_h$-smooth, and $S_h$-Hessian smooth in $(x, y)$. That is, $\|\nabla h(x, y)\| \le L_h$, $\forall i : \|\nabla^2 h_i(x, y)\| \le C_h$, $\|\nabla^2 h_i(x, y) - \nabla^2 h_i(\bar{x}, \bar{y})\| \le S_h \|(x, y) - (\bar{x}, \bar{y})\|$.

3. For every $x$, the LL primal and dual solution $(y^*, \lambda^*)$ and its corresponding active set $\mathcal{I}$ satisfy $\|\nabla_y h_{\mathcal{I}}(x, y^*)^{\dagger}\| \le C_B$, $\|y^*\| \le R_y$, $\|\lambda^*\|_1 \le R_\lambda$.

We make the first assumption because, for general convex constraints, we are not aware of any algorithm that achieves linear convergence for the *dual* solution. To draw an analogy with the linear-constrained case, we replace a concrete algorithm with an oracle and focus on the oracle complexity rather than the gradient complexity. The second assumption is mild and analogous to those for the objective. Although the third assumption is somewhat technical, similar assumptions have been commonly used in bilevel literature, even for linear constraints. With these assumptions in place, we are ready to extend the guarantees from the linear constraint setting to the broader class of well-behaved convex constraints.

## D.2. Proofs and Technical Details

For general convex constraints, consider the *ghost* lower-level problem expanded at approximate primal and dual solutions $(\tilde{y}, \tilde{\lambda})$. Let $\mathsf{B} = \nabla_y h_{\mathcal{I}}(x, y^*)$ and $b = \nabla_y h_{\mathcal{I}}(x, y^*) y^*$. Similarly, define $\tilde{\mathsf{B}} = \nabla_y h_{\mathcal{I}}(x, \tilde{y})$ and $\tilde{b} = \nabla_y h_{\mathcal{I}}(x, \tilde{y}) \tilde{y}$. We have $\|\mathsf{B} - \tilde{\mathsf{B}}\| \le C_h \|y^* - \tilde{y}\|$, and $\|b - \tilde{b}\| \le \|\mathsf{B} - \tilde{\mathsf{B}}\| \|y^*\| + \|\tilde{\mathsf{B}}\| \|y^* - \tilde{y}\| \le (C_h R_y + L_h) \|y^* - \tilde{y}\|$. We next show that $y^* = \arg\min_{\mathsf{B}y=b} g(x, y) + \lambda^{*\top} h(x, y)$ and $\tilde{\tilde{y}} = \arg\min_{\tilde{\mathsf{B}}y=\tilde{b}} g(x, y) + \tilde{\lambda}^\top h(x, y)$ are close. Let us introduce $y^+ = \arg\min_{\tilde{\mathsf{B}}y=\tilde{b}} g(x, y) + \lambda^{*\top} h(x, y)$.

*Lemma* D.2. *Assume that* $\|y^* - \tilde{y}\| \le \min\{\epsilon, \frac{C_B}{2C_h}\}$, *we have* $\|y^* - y^+\| \le \mathcal{O}(\epsilon)$.

*Proof.* By Weyl's inequality for singular values, $|\sigma_1(\mathsf{B}) - \sigma_1(\tilde{\mathsf{B}})| \le C_h \|y^* - \tilde{y}\| \le C_B/2$. Therefore, $\|\tilde{\mathsf{B}}^{\dagger}\| = 1/\sigma_1(\tilde{\mathsf{B}}) \le 2/C_B$. Denote by $y_{\tilde{\mathsf{B}}}^*$ the Euclidean projection of $y^*$ onto $\tilde{\mathsf{B}}y = \tilde{b}$. By simple linear algebra, $y_{\tilde{\mathsf{B}}}^* = y^* - \tilde{\mathsf{B}}^{\dagger}(\tilde{\mathsf{B}}y^* - \tilde{b})$ and thus $\|y^* - y_{\tilde{\mathsf{B}}}^*\| \le \|\tilde{\mathsf{B}}^{\dagger}\| \|\tilde{\mathsf{B}}y^* - \tilde{b}\|$. Substituting the bounds, we have $\|\tilde{\mathsf{B}}^{\dagger}\| \le 2/C_B$ and $\|\tilde{\mathsf{B}}y^* - \tilde{b}\| = \|(\tilde{\mathsf{B}} - \mathsf{B})y^* - (\tilde{b} - b)\| \le C_h R_y \epsilon + (C_h R_y + L_h)\epsilon$. In short, we obtain $\|y^* - y_{\tilde{\mathsf{B}}}^*\| \le \mathcal{O}(\epsilon)$.

Note that $y^*$ is in fact the global optimum of $g(x, y) + \lambda^{*\top} h(x, y)$. Hence,

$$
\begin{aligned}
g(x, y^+) + \lambda^{*\top} h(x, y^+) &\le g(x, y_{\tilde{\mathsf{B}}}^*) + \lambda^{*\top} h(x, y_{\tilde{\mathsf{B}}}^*) \\
&\le g(x, y^*) + \lambda^{*\top} h(x, y^*) + \frac{C_g + C_h R_\lambda}{2} \|y^* - y_{\tilde{\mathsf{B}}}^*\|^2 \\
&\le g(x, y^+) + \lambda^{*\top} h(x, y^+) - \frac{\mu_g}{2} \|y^* - y^+\|^2 + \frac{C_g + C_h R_\lambda}{2} \|y^* - y_{\tilde{\mathsf{B}}}^*\|^2
\end{aligned}
$$

Therefore, we get $\|y^* - y^+\| \le \sqrt{\frac{C_g + C_h R_\lambda}{\mu_g}} \|y^* - y_{\tilde{\mathsf{B}}}^*\| \le \mathcal{O}(\epsilon)$. $\qquad \square$

*Lemma* D.3. *Assume that* $\|\lambda^* - \tilde{\lambda}\| \le \epsilon$, *we have* $\|y^+ - \tilde{\tilde{y}}\| \le \mathcal{O}(\epsilon)$.

*Proof.* By the optimality of $y^+$,

$$g(x, y^+) + \lambda^{*\top} h(x, y^+) + \frac{\mu_g}{2}\|y^+ - \tilde{\tilde{y}}\|^2 \le g(x, \tilde{\tilde{y}}) + \lambda^{*\top} h(x, \tilde{\tilde{y}}) \ .$$

Similarly, we have

$$g(x, \tilde{\tilde{y}}) + \tilde{\lambda}^\top h(x, \tilde{\tilde{y}}) + \frac{\mu_g}{2}\|y^+ - \tilde{\tilde{y}}\|^2 \le g(x, y^+) + \tilde{\lambda}^\top h(x, y^+) \ .$$

Summing up the two inequalities we get,

$$\mu_g\|y^+ - \tilde{\tilde{y}}\|^2 \le (\tilde{\lambda} - \lambda^*)^\top (h(x, y^+) - h(x, \tilde{\tilde{y}})) \le \epsilon L_h \|y^+ - \tilde{\tilde{y}}\| \ .$$

Therefore, $\|y^+ - \tilde{\tilde{y}}\| \le \frac{L_h \epsilon}{\mu_g}$. $\qquad \square$

Combining Lemma D.2 and Lemma D.3, we get $\|y^* - \tilde{\tilde{y}}\| \le \mathcal{O}(\epsilon)$. With the solutions being close, we next show that $\left\|\frac{dy^*}{dx} - \frac{d\tilde{\tilde{y}}}{dx}\right\| \le \mathcal{O}(\epsilon)$. We slightly abuse the notation to denote $G = \nabla_{yy} g(x, y^*) + \lambda^\top \nabla_{yy} h(x, y^*)$ and $\tilde{G} = \nabla_{yy} g(x, \tilde{\tilde{y}}) + \tilde{\lambda}^\top \nabla_{yy} h(x, \tilde{\tilde{y}})$. From Lemma A.1, we have

$$\frac{dy^*}{dx} = \Pi_{\mathsf{B}}^G\Big(-G^{-1}\big(\nabla_{yx} g(x, y^*) + \lambda^{*\top} \nabla_{yx} h(x, y^*)\big)\Big) + (\mathsf{I} - \Pi_{\mathsf{B}}^G)(\mathsf{B}^\dagger \mathsf{A}) \tag{31}$$

$$\frac{d\tilde{\tilde{y}}}{dx} = \Pi_{\tilde{\mathsf{B}}}^{\tilde{G}}\Big(-\tilde{G}^{-1}\big(\nabla_{yx} g(x, \tilde{\tilde{y}}) + \tilde{\lambda}^\top \nabla_{yx} h(x, \tilde{\tilde{y}})\big)\Big) + (\mathsf{I} - \Pi_{\tilde{\mathsf{B}}}^{\tilde{G}})(\tilde{\mathsf{B}}^\dagger \tilde{A}) \ . \tag{32}$$

*Lemma* D.4. *Given* $\mu_g \mathsf{I} \preceq G \preceq C_g \mathsf{I}$, $\|\mathsf{B} - \tilde{\mathsf{B}}\| \le \epsilon$, $\max\{\|\mathsf{B}\|, \|\tilde{\mathsf{B}}\|\} \le C_h$ *and* $\max\{\|\mathsf{B}^\dagger\|, \|\tilde{\mathsf{B}}^\dagger\|\} \le C_B$, *we have* $\|\Pi_{\mathsf{B}}^G - \Pi_{\tilde{\mathsf{B}}}^G\| \le \mathcal{O}(\epsilon)$.

*Proof.* By reparameterization, $\|\Pi_{\mathsf{B}}^G - \Pi_{\tilde{\mathsf{B}}}^G\| = \|G^{-\frac{1}{2}}\big(\Pi_{\mathsf{B}G^{-\frac{1}{2}}} - \Pi_{\tilde{\mathsf{B}}G^{-\frac{1}{2}}}\big)G^{\frac{1}{2}}\| \le \sqrt{\frac{C_g}{\mu_g}} \cdot \|\Pi_{\mathsf{B}G^{-\frac{1}{2}}} - \Pi_{\tilde{\mathsf{B}}G^{-\frac{1}{2}}}\|$. For the Euclidean projection, it is known that

$$\begin{aligned}\|\Pi_{\mathsf{B}G^{-\frac{1}{2}}} - \Pi_{\tilde{\mathsf{B}}G^{-\frac{1}{2}}}\| &= \|(\mathsf{B}G^{-\frac{1}{2}})^\dagger \mathsf{B}G^{-\frac{1}{2}} - (\tilde{\mathsf{B}}G^{-\frac{1}{2}})^\dagger \tilde{\mathsf{B}}G^{-\frac{1}{2}}\| \\ &\le \sqrt{\frac{1}{\mu_g}} \cdot \Big\{\|(\mathsf{B}G^{-\frac{1}{2}})^\dagger(\mathsf{B} - \tilde{\mathsf{B}})\| + \|((\mathsf{B}G^{-\frac{1}{2}})^\dagger - (\tilde{\mathsf{B}}G^{-\frac{1}{2}})^\dagger)\tilde{\mathsf{B}}\|\Big\}\end{aligned}$$

where from perturbation theory (e.g. (Stewart, 1977)),

$$\big\|(\mathsf{B}G^{-\frac{1}{2}})^\dagger - (\tilde{\mathsf{B}}G^{-\frac{1}{2}})^\dagger\big\| \le \sqrt{2}\|(\mathsf{B}G^{-\frac{1}{2}})^\dagger\|\|(\tilde{\mathsf{B}}G^{-\frac{1}{2}})^\dagger\|\|(\mathsf{B} - \tilde{\mathsf{B}})G^{-\frac{1}{2}}\|$$

In addition, we have $\|(\mathsf{B}G^{-\frac{1}{2}})^\dagger\| \le C_B \sqrt{C_g}$. Putting everything together, we get

$$\|\Pi_{\mathsf{B}}^G - \Pi_{\tilde{\mathsf{B}}}^G\| \le \sqrt{\frac{C_g}{\mu_g}}\sqrt{\frac{1}{\mu_g}}\{C_B\sqrt{C_g}\epsilon + \sqrt{2}C_B^2 C_g \epsilon \mu_g^{-1/2} C_h\} \le \mathcal{O}(\epsilon) \ .$$

$\qquad \square$

The remaining proof can be completed by comparing Appendix D.2 and Appendix D.2, where all the appearing terms are bounded by $\mathcal{O}(1)$ and the differences are bounded by $\mathcal{O}(\epsilon)$.

*Corollary* D.5. *Under the assumptions made for general convex constraints, there exists an algorithm, which in* $\tilde{\mathcal{O}}(\delta^{-1}\epsilon^{-3})$ *oracle calls converges to a* $(\delta, \epsilon)$-*stationary point for the bilevel problem with general convex constraints.*

*Proof.* We have shown that both the solution and the hypergradient of the ghost problem, constructed from an approximate solution, are close to those of the original problem. Consequently, the same guarantees that hold in the linear case apply here as well. $\qquad \square$

## E. Proof for Solver-agnostic Reformulation

*Lemma* E.1 (Solver-Agnostic reformulation). *Fix $x$ and let us define $c := detach(\nabla_y f(x, y^*(x)), \hat{f}(x, y) := \langle c, y \rangle$. Let $(\hat{y}^*_\delta(x), \hat{\lambda}^*_\delta(x))$ be the primal–dual solution of the perturbed ghost problem Problem P3 with perturbation term $\delta f(x, y)$ replaced by $\delta \hat{f}(x, y) = \delta \langle c, y \rangle$, and define the corresponding finite-difference estimator (cf. Equation 4)*

$$\hat{v}_x(\delta) := \frac{1}{\delta} \left( \nabla_x \big[ \tilde{g}(x, \hat{y}^*_\delta) + \langle \hat{\lambda}^*_\delta, \tilde{h}(x, \tilde{y}^*) \rangle \big] - \nabla_x \big[ \tilde{g}(x, \tilde{y}^*) + \langle \tilde{\lambda}, \tilde{h}(x, \tilde{y}^*) \rangle \big] \right). \tag{33}$$

*Under the assumptions of Lemma Lemma C.1, there exists $C > 0$ such that for all sufficiently small $\delta$,*

$$\left\| \hat{v}_x(\delta) - \left( \frac{dy^*(x)}{dx} \right)^\top \nabla_y f(x, y^*(x)) \right\| \leq C\,\delta. \tag{34}$$

*Proof.* By construction, $\nabla_y \hat{f}(x, y) = c$ for all $(x, y)$, hence at the fixed point $x$,

$$\nabla_y \hat{f}(x, \tilde{y}^*(x)) = c = \nabla_y f(x, y^*(x)),$$

where we used $\tilde{y}^*(x) = y^*(x)$ (cf. Theorem B.1). Applying Lemma Lemma C.1 to the perturbed ghost problem with perturbation function $\hat{f}$ gives

$$\left\| \hat{v}_x(\delta) - \left( \frac{d\tilde{y}^*(x)}{dx} \right)^\top \nabla_y \hat{f}(x, \tilde{y}^*(x)) \right\| \leq C\,\delta.$$

Using $\frac{d\tilde{y}^*(x)}{dx} = \frac{dy^*(x)}{dx}$ and $\tilde{y}^*(x) = y^*(x)$ at $x$ (cf. Theorem B.1) yields the desired bound. $\qquad\square$

## F. Discussion for Active Set Requirement

We now argue the necessity of the active set requirement. Suppose one only has access to an oracle that guarantees an approximate solution $(\tilde{y}, \tilde{\lambda})$ such that $\|\tilde{y} - y^*\| + \|\tilde{\lambda} - \lambda^*\| \leq \epsilon$. Consider the one-dimensional problem with $g(x, y) = \frac{1}{2}(y - a)^2$ and $h(x, y) = y - ax$ for some $a > 1$ (say, $a = 100$). For $x = 1 - \frac{\epsilon}{2a}$ we have $y^* = a - \frac{\epsilon}{2}$, $\lambda^* = \frac{\epsilon}{2}$ and $y'(x) = a$. In contrast, for $\bar{x} = 1 + \frac{\epsilon}{2a}$ we have $y^* = a$, $\lambda^* = 0$, and $y'(\bar{x}) = 0$. However, in both cases $(\tilde{y}, \tilde{\lambda}) = (a, 0)$ is a valid oracle output. Moreover, $|x - \bar{x}| = \frac{\epsilon}{a}$. Thus, no continuous function of $(x, \tilde{y}, \tilde{\lambda})$ can approximate the gradient accurately in both cases. This is formalized in Lemma F.1.

We further complement the result by showing that a slightly smoothed version of the problem can be solved efficiently without requiring an active set assumption.

*Lemma* F.1. *No continuous estimator $\hat{\nabla} F$ can guarantee $\|\hat{\nabla} F(x) - \nabla F(x)\| \leq o(1) \; \forall \; x \in \mathcal{X}$.*

*Proof.* For any $a > 1$ and $\epsilon > 0$, consider the example given above and let $f(x, y) = y$. We have for any $x \in (1 - \frac{\epsilon}{2a}, 1)$ : $\nabla F(x) = a$ ; and for $x \in (1, 1 + \frac{\epsilon}{2a})$ : $\nabla F(x) = 0$. Assume $\hat{\nabla} F(1) > \frac{a}{2}$. In this case, $\lim_{x \to 1^+} |\hat{\nabla} F(x) - \nabla F(x)| > \frac{a}{2}$. Assume $\hat{\nabla} F(1) < \frac{a}{2}$. In this case, $\lim_{x \to 1^-} |\hat{\nabla} F(x) - \nabla F(x)| > \frac{a}{2}$. $\qquad\square$

*Theorem* F.2. *Assume that $\forall x \in \mathcal{X}$, the radius of the largest ball contained in $\mathsf{A}x - \mathsf{B}y - b \leq 0$ is bounded below by $\bar{\rho}$. $\forall \rho < \bar{\rho}$, define $\bar{F}_\rho(x) = \mathbb{E}_{\eta \sim U(-\rho, \rho)^m} \big[ f(x, y_\eta(x)) \big]$, where $y_\eta(x) = \arg\min_{\mathsf{A}x - \mathsf{B}y - b \leq \eta} g(x, y)$. There exists an algorithm that does not require the active set assumption and outputs a point $x^{out}$ such that $\mathbb{E} \big[ dist(0, \partial_\delta \bar{F}_\rho(x^{out}) \big] \leq \epsilon + \alpha$ with $T = \mathcal{O}\left( \frac{1}{\delta \epsilon^3} \log\left( \frac{m}{\rho \alpha} \right) \right)$ oracle calls to $f$ and $g$.*

*Proof.* For a given $\rho$, we construct the following stochastic gradient estimator $\tilde{g}$ by first sampling $\eta \sim U(-\rho, \rho)^m$ and then applying Algorithm 1 to the perturbed constraints. Assume w.l.o.g., that the 2-norm of each row of $\mathsf{B}$ is normalized to 1. In the following, we show one can choose the tolerance $\varepsilon$ of the algorithm small enough such that, with high probability, our gradient estimator is good. Fix an $x$, and let $\tilde{b} = \eta + b - \mathsf{A}x$. Observe that if $y^*$ and $\lambda^*$ satisfy $\mathsf{B}y^* - \tilde{b} \notin [-\varepsilon, 0)$ and $\lambda^* \notin (0, \varepsilon]$, no ambiguity can occur and thus we can guarantee an approximation of $\mathcal{O}(\varepsilon)$. In addition, since the ground truth and our estimator are both bounded by $\mathcal{O}(1)$, we obtain a guarantee of the form $\mathbb{E}\|\tilde{g} - g\| \leq \mathcal{O}(\alpha)$ and $\mathbb{E}\|\tilde{g}\|^2 \leq \mathcal{O}(1)$, as in Lemma B.2 of (Kornowski et al., 2024). Therefore, the results follow.

For $i \in \{1, \ldots, m\}$, let $E_i$ be the bad event $B_i^\top y^* - \tilde{b}_i \in [-\epsilon, 0) \vee \lambda_i^* \in [0, \varepsilon)$. Fix the rest coordinates of $\tilde{b}$, and let $g_i(y) = g(y) + \mathbb{1}\{B_{-i} y \leq \tilde{b}_{-i}\}$. Then we have $y^*(\eta_i) = \arg\min_y g_i(y) + \lambda_i^*(\eta_i) B_i^\top y$. For the first part (i.e., when the constraint is not active), $\eta_i$ must lie in an interval of length $\varepsilon$.

For the second case, consider two values $\eta_i, \eta_i'$. From the optimality conditions and the strong convexity of $g_i$, we have

$$\frac{\mu}{2}\|y^*(\eta_i) - y^*(\eta_i')\|^2 + g_i(y^*(\eta_i)) + \lambda_i^*(\eta_i) B_i^\top y^*(\eta_i) \quad \leq \quad g_i(y^*(\eta_i')) + \lambda_i^*(\eta_i) B_i^\top y^*(\eta_i')$$

$$\frac{\mu}{2}\|y^*(\eta_i) - y^*(\eta_i')\|^2 + g_i(y^*(\eta_i')) + \lambda_i^*(\eta_i') B_i^\top y^*(\eta_i') \quad \leq \quad g_i(y^*(\eta_i)) + \lambda_i^*(\eta_i') B_i^\top y^*(\eta_i)$$

Summing the two inequalities gives

$$\mu\|y^*(\eta_i) - y^*(\eta_i')\|^2 \leq (\lambda_i^*(\eta_i) - \lambda_i^*(\eta_i')) B_i^\top (y^*(\eta_i) - y^*(\eta_i')) = (\lambda_i^*(\eta_i) - \lambda_i^*(\eta_i'))(\tilde{b}_i(\eta_i') - \tilde{b}_i(\eta_i)) \ .$$

In addition, we have

$$\|y^*(\eta_i) - y^*(\eta_i')\| \geq |B_i^\top (y^*(\eta_i) - y^*(\eta_i'))| = |\tilde{b}_i(\eta_i') - \tilde{b}_i(\eta_i)| \ .$$

Therefore, $\frac{1}{\mu}|\lambda_i^*(\eta_i) - \lambda_i^*(\eta_i')| \geq |\tilde{b}_i(\eta_i') - \tilde{b}_i(\eta_i)| = |\eta_i' - \eta_i|$, which implies that $\eta_i$ must lie in an interval of length $\frac{\varepsilon}{\mu}$.

Apply the union bound, $\mathbf{P}(\bigvee_{i=1}^m E_i) \leq \mathcal{O}(\frac{m\varepsilon}{\rho})$. Setting $\varepsilon = \mathcal{O}(\frac{\rho\alpha}{m})$ yields the desired result. $\qquad \square$

**Linearized active cone constraint**  Previously, we discussed that for scalar inequality, the active set reduction means keeping the active manifold and linearizing it as equalities. Here, we assume that we have cone constraints. So we need to linearize the active manifold by replacing the curved surface with its tangent plane. We can define the following indicator function $\mathbb{I} : \mathcal{X} \to \mathbb{R}$:

$$\mathbb{I}_{\mathcal{K}}(x) = \mathbb{I}\{x \in \mathcal{K}\} := \begin{cases} 0, & \text{if } x \in \mathcal{K} \\ \infty, & \text{if } x \notin \mathcal{K} \end{cases} . \tag{35}$$

At a feasible optimum $x^*$, we have $x \in \mathcal{K}$ and thus $\mathbb{I}_{\mathcal{K}}(x^*) = 0$. Let $N_{\mathcal{K}}(x) := \{d \in \mathbb{R}^n \mid \forall y \in \mathcal{K}, d^\top x \geq d^\top y\}$ denote the normal cone of $\mathcal{K}$ at point $x$. We know if $x \in \mathcal{K}$, then $\partial \mathbb{I}_{\mathcal{K}} = N_{\mathcal{K}}(x)$. At $x^*$, there exists a Lagrange Multiplier $\lambda^* \in -\mathcal{K}^*$, such that $\lambda^* \in \partial \mathbb{I}_{\mathcal{K}}$. The local linearization can be computed as

$$\mathbb{I}_{\mathcal{K}}(x) \approx \mathbb{I}_{\mathcal{K}}(x^*) + \partial_x \mathbb{I}_{\mathcal{K}}(x^*)^\top (x - x^*). \tag{36}$$

Since $\lambda^* \in \partial_z \mathbb{I}_{\mathcal{K}}(x)$, we know

$$\mathbb{I}_{\mathcal{K}}(x) \approx (\lambda^*)^\top (x - x^*). \tag{37}$$

We can rewrite the ghost problem as

$$\min_{y,t} g(x, y) + \lambda^* h_C(x, y) \text{ s.t. } \nabla_y h(x, y)(y - y^*) = 0 \tag{38}$$

# G. Experiment Setup and Results

## G.1. Experiment Setup

We repeat each experiment with 10 random seeds. For synthetic tasks, we generate a dataset of 2048 samples with 80% for training and 20% for testing. To ensure a fair comparison, all methods use the same hyperparameters: we train using the Adam optimizer with a fixed learning rate and batch size across methods. We train for a sufficient number of epochs such that all methods converge. The neural network models used in each task are simple 2-layer MLPs.

In the synthetic tasks, the input dimension is $d_x = 640$ and the output dimension $d_y = 800$. We set solver tolerances $\epsilon$ to balance accuracy and speed: $\epsilon = 10^{-6}$ for the synthetic DFL task, and $\epsilon = 10^{-4}$ for SOCP. Notably, we find that the `LPGD` requires a much tighter tolerance than default; $10^{-12}$ instead of $10^{-4}$ to converge in some tasks, as discussed later.

For the compute environment, all runs were executed on a Slurm server, allocating 8 CPU cores and sufficient memory to avoid out-of-memory failures (48GB per core, total 384GB). We keep the compute allocation fixed across methods and seeds to ensure fair runtime comparisons.

We also implement `Alt-Diff` (Sun et al., 2023) as one of our baselines. But similar to previous work (e.g., Table 2 in (Bambade et al., 2024)), we find it is too slow. So we do not include this baseline in the paper.

## G.2. Additional Ablation Studies

In this section, we perform a series of ablation studies to better understand the performance and stability of our proposed method.

**Memory**  Figure 7 compares peak GPU memory as we scale the lower-level dimension $d_y$ and reports memory cost for initialization (solver/canonicalization setup), forward (solving the lower-level problem), and backward (hypergradient computation). It indicates that `FFOCP` stays flat, indicating that its first-order oracle can be executed with a small, stable memory cost. In contrast, `CvxpyLayer`'s memory usage grows rapidly and becomes prohibitive for large $d_y$, indicating that the costs of canonicalization and implicit differentiation are substantial. This gap highlights a key advantage of `FFOCP`: it avoids storing large factorization/intermediate solver state and computes hypergradients using only first-order information, making it more scalable in memory.

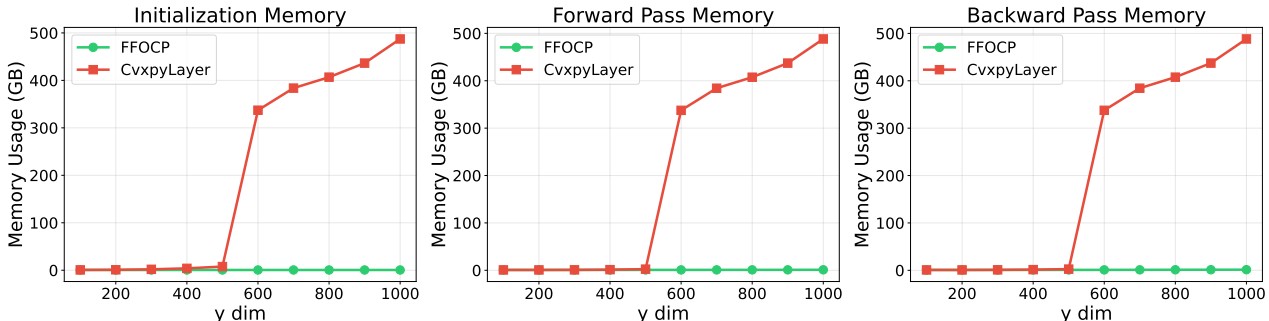

*Figure 7.* Memory usage vs. lower-level decision dimension $d_y$ on the synthetic DFL task. We report peak GPU memory (GD) for the initialization, the forward pass, and the backward pass. `CvxpyLayer` shows a sharp increase in memory as $d_y$ grows, while `FFOCP` remains nearly constant across all three stages.

**LPGD comparison**  As mentioned in Section 7, we discuss that the default solver tolerance is insufficient for `LPGD` to converge in Sudoku and synthetic DFL tasks. As shown in Figure 8, `LPGD` with the default tolerance exhibits unstable training loss, while tightening the solver tolerance to $\epsilon = 10^{-12}$ restores convergence. However, Figure 8 (middle/right) shows that this accuracy comes at a substantial computation cost, suggesting that `LPGD` requires very accurate inner solves in our setup—likely explaining its higher runtimes compared with the results reported in Paulus et al. (2024).

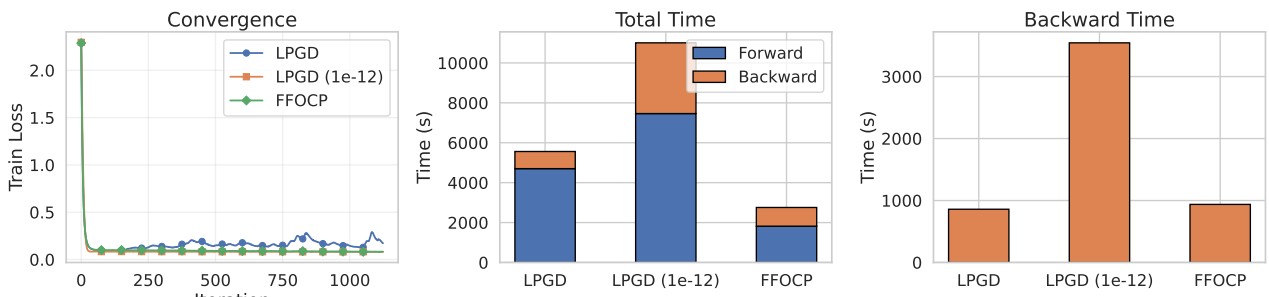

*Figure 8.* Compare the performance of `LPGD` with its default tolerance, `LPGD` with a tightened tolerance ($\epsilon = 10^{-12}$), and our method `FFOCP` in the Sudoku task. Left: convergence performance in Sudoku. Middle: total wall-clock with forward and backward times. Right: wall-clock for backward times. Tightening `LPGD`'s tolerance restores stable convergence, but dramatically increases runtime, primarily due to a much more expensive backward pass, whereas `FFOCP` remains stable with lower overall cost.

**Compare with `QPTH` with GPU**  In the main paper, we report the results for `QPTH` with CPU. Here, we also compare with `QPTH` with GPU. While `QPTH` with GPU substantially speeds up the forward solve compared to `QPTH` with CPU, it does not consistently improve end-to-end training time. In Figure 9, `FFOQP` matches `QPTH`'s convergence on both the synthetic QP and Sudoku tasks, and is competitive in total time: depending on the task, `FFOQP` can be faster than `QPTH` with GPU.

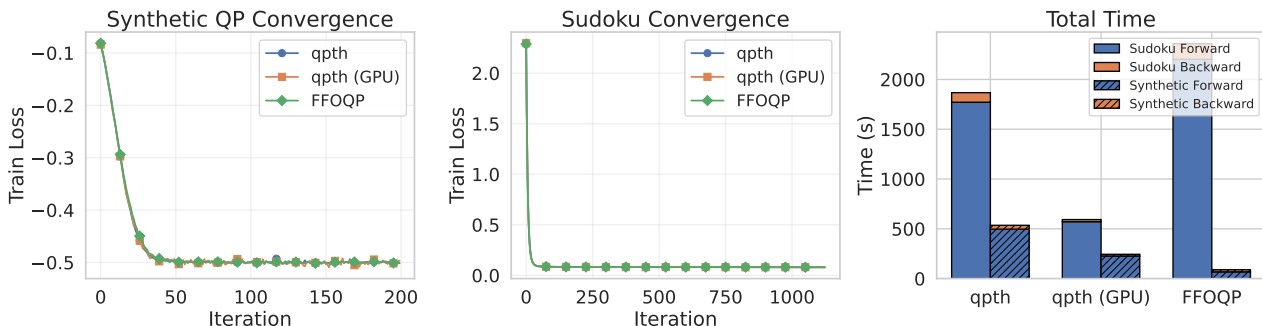

*Figure 9.* Compare the performance of QPTH with CPU, QPTH with GPU, and our method FFOQP. Left/middle: convergence performance in synthetic QP and Sudoku.

**Additional results for the scalability with problem size** In Section 7, we present the results for the scalability with problem size for the QP task. In Figure 10, we present the results for the SOCP task.

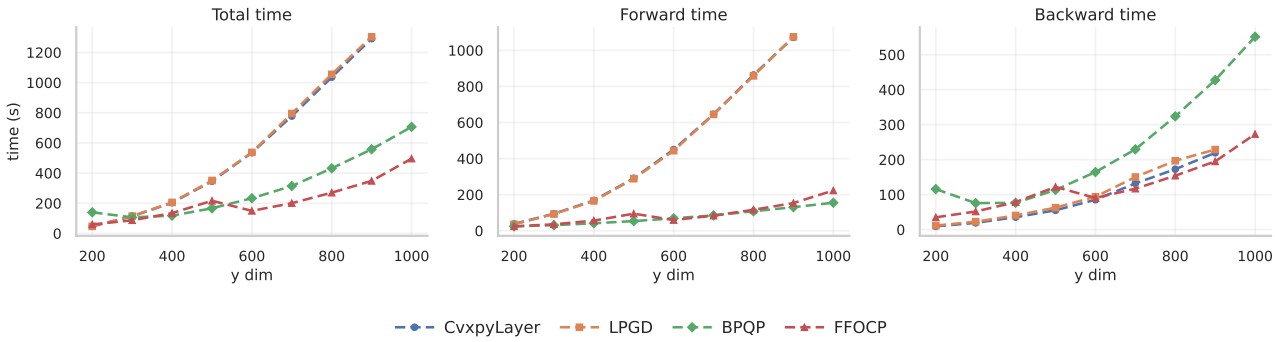

*Figure 10.* Different variable dimension for the SOCP task.

**Gradient distance and similarity** We examine the accuracy of our approximated hypergradient by comparing it with the ground-truth hypergradient produced by CvxpyLayer[2]. We then compute the cosine similarity between our FFO gradient and the reference gradient, as well as the $\ell_2$ and $\ell_\infty$ norm differences. As shown in Figure 11, we can observe that the cosine similarity is consistently high at the beginning of the training process, where the gradient norm is high. In the meantime, the small magnitude of the error suggests that any bias introduced by our approximation is minor. These results provide practical validation of our oracle's accuracy: even though we use an approximate dual solution, the active set identification is relatively correct, such that the gradient direction is accurate.

**Batch size** We study the effect of batch size on throughput and efficiency. In a typical end-to-end training scenario, solving multiple optimization instances in parallel (batch-solving) can amortize overhead costs. We measure the throughput, the time per training step, and the per-sample cost as the batch size increases, as shown in Figure 12. It shows that batch-solving substantially improves efficiency. As batch size increases, throughput rises while time per step drops quickly and then levels off, indicating that fixed per-step overhead is amortized and the runtime becomes compute-dominated. Therefore, the per-sample cost decreases by orders of magnitude at larger batches.

**Solver tolerance** We also analyze how sensitive the end-to-end training performance is to the choice of solver tolerance for solving the perturbed problem (Problem P3). A very strict tolerance (e.g., $10^{-12}$) means the perturbed solution $y_\delta$ is found with high precision, which could improve gradient accuracy but will take more solver iterations, whereas a looser tolerance (e.g., $10^{-3}$) speeds up each solve but yields a less precise solution. In our FFO approach, thanks to the allowance of an approximate dual, we expect that we do not require extremely tight tolerances to maintain training stability. The

---

[2]Since an exact hypergradient oracle is generally intractable for large problems, we consider CvxpyLayer with high tolerance as a proxy oracle to produce ground-truth hypergradients.

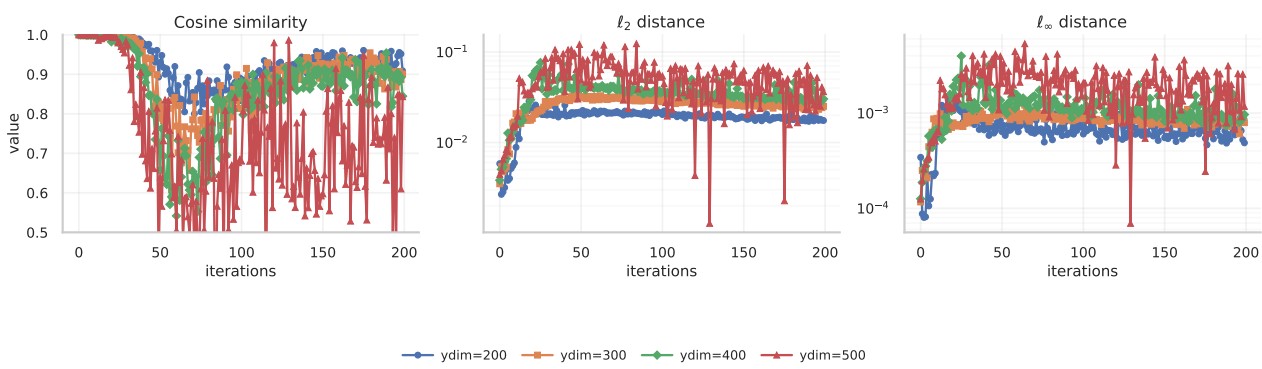

*Figure 11.* Gradient cosine similarity between `FFOCP` and `CvxpyLayer` for different variable dimensions.

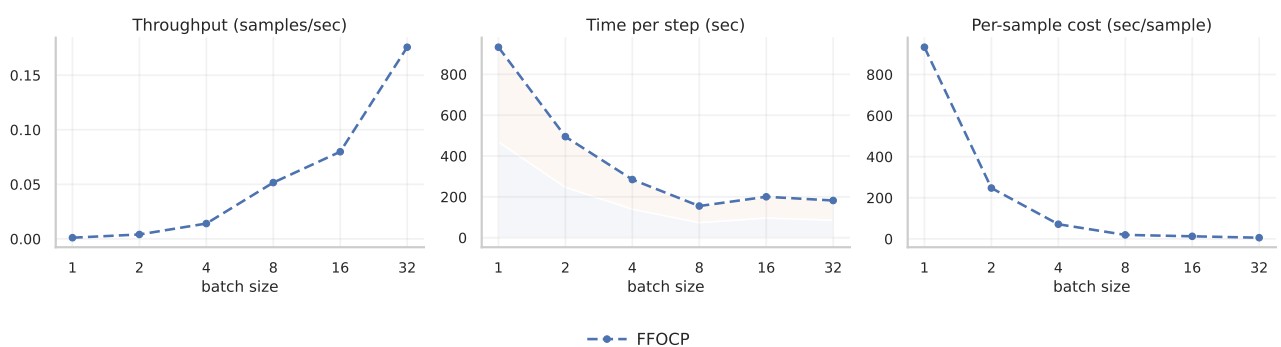

*Figure 12.* Throughput and time per step for different batch sizes.

ablation experiments support this expectation. With higher (looser) tolerance, Figure Figure 13 shows that the backward time of our method FFOCP is faster without affecting convergence of the loss.

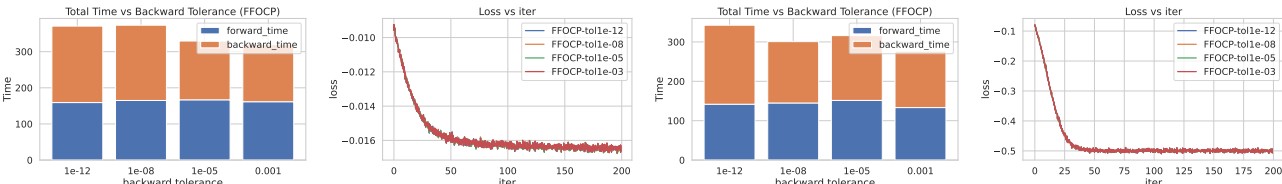

*Figure 13.* Computation costs and losses for different backward solving tolerances in the DFL QP task (first two figures) and SOCP task (last two figures).

### G.3. Additional related works

**Bilevel Optimization with Non-Convex Lower-Level Objectives** . Bilevel Optimization (BO) has a long history in operations research, where the lower-level problem serves as the constraint of the upper-level problem (Von Stackelberg et al., 1953; Bracken & McGill, 1973; Ye & Zhu, 1995). Currently, it has been widely utilized in a large number of practical applications, such as hyperparameter optimization (Feurer & Hutter, 2019), meta learning (Finn et al., 2017; Rajeswaran et al., 2019), adversarially robust learning (Bai et al., 2021), data reweighting (Holstege et al., 2024), neural architecture search (Elsken et al., 2019), speech recognition (Saif et al., 2024), and more recently, the pretraining-finetuning pipeline for large models (Shen et al., 2024).

BO with non-convex lower-level objectives poses significant computational challenges and is generally intractable without additional assumptions, even in simplified settings such as composite optimization (Danilova et al., 2022) and min-max optimization (Razaviyayn et al., 2020). To address this, recent approaches have incorporated assumptions such as local Lipschitz continuity (Chen et al., 2024) and the PL condition (Shen & Chen, 2023), to ensure theoretical traceability.

Furthermore, Kwon et al. (Kwon et al., 2024b) studied the minimal assumption of the continuity of lower-level solution sets, and guaranteed the gradient-wise asymptotic equivalence. Following this line, many other approaches have been proposed (Liu et al., 2022; 2021a;b; Arjevani et al., 2023; Huang, 2024; Chen et al., 2024; Chu et al., 2024; Liu et al., 2024; Yao et al., 2024).

In particular, for non-convex neural networks, the convergence analysis remains non-trivial, and performance guarantees within BO contexts are still missing. Franceschi et al. (Franceschi et al., 2018) target on analyzing the convergence of BO in hyperparameter optimization and meta learning by assuming the inner-level solution set is a singleton. To our best knowledge, only two theoretical works have considered the case where the lower-level objective involves neural network training, focusing on the non-singleton issue caused by multiple global minima of neural network training. Arbel et al. (Arbel & Mairal, 2022) proposed a selection-map algorithm based on gradient flows, and proved that it can approximately solve the non-singleton BO up to a bias. Furthermore, Petrulionyte et al. (Petrulionyte et al., 2024) introduced a novel approach by transforming the strong convexity assumption of network parameters into functional vectors, which represents a milder condition. And they further proposed a single-loop algorithm with a warm-start mechanism to implement the functional implicit differentiation effectively.

There are already lots of work focusing on analyzing the convergence and generalization performance of neural networks in hyperparameter optimization (Bao et al., 2021). However, most of them are analyzing the gradient unrolling or implicit bias algorithms, rather than fully first-order algorithms.

**Efficiently Computing the Optimization Hypergradient**  The main algorithmic challenge in differentiable optimization is computing the hypergradients of the model loss with respect to the inputs of the optimization problem. Importantly, the implicit derivatives that arise in differentiable layers, hyperparameter tuning, and bilevel learning are mathematically identical: each requires differentiating the lower-level KKT system to obtain a hypergradient. Two main families of estimators are used in practice. *Implicit differentiation* methods (Krantz & Parks, 2002; Domke, 2012; Maclaurin et al., 2015a) linearize the KKT system and solve the resulting linear equations either with direct factorizations of the KKT/Hessian matrix (e.g., the approach used in Amos & Kolter (2017)) or with iterative routines such as conjugate gradients (Xue et al., 2021) and Neumann series (Ghadimi & Wang, 2018), as done in Agrawal et al. (2019a). *Gradient unrolling* (GU) approaches (Maclaurin et al., 2015b), such as Pineda et al. (2022), backpropagate through a truncated execution of the inner solver and therefore only require forward-mode gradients produced by automatic differentiation. While unrolling avoids explicit Hessian solves, its memory usage grows linearly with the number of iterations, and it can suffer from truncation bias. Recent work has therefore focused on reducing the cost of both strategies—via custom backward passes (Paulus et al., 2024; Pan et al., 2024; Sun et al., 2023), checkpointing, or low-rank updates—yet these methods still depend on either second-order information or storing long optimization trajectories, which limits their scalability.

