# OpenReview forum: "A Fully First-Order Layer for Differentiable Optimization"
_ICML.cc/2026/Conference — ICML 2026 spotlight_

### Official Review · Reviewer_RMnH · 2026-03-04

**Soundness:** 3
**Presentation:** 3
**Significance:** 3
**Originality:** 3
**Overall Recommendation:** 5
**Confidence:** 2

**Summary:**

Standard approaches in differentiable optimisation generally relies on implicit differentiation, and thus require the computation of a Hessian which is generally very expensive. This paper proposes to formulate differentiable optimisation as an instance of a bi-level optimization problem that is solved by using only first order information (that is from gradient and objective function evaluations). A theoretical study is proposed of the overall complexity of this procedure. Various experimental results are also reported that show performances similar to existing procedures while improving scalability as the size of the problem increases. The method is finally implemented in a Python library that is compatible with existing differentible solvers.

**Compliance With Llm Reviewing Policy:**

Affirmed.

**Final Justification:**

I kept my accept score and had no further questions for the reviewers, my opinion remains unchanged after the rebuttal phase.

**Key Questions For Authors:**

No further questions.

**Limitations:**

yes

**Strengths And Weaknesses:**

* Soundness: The paper is  technically sound and rigorous with an extensive comparison to the literature on differentiable optimization both theoretically and empirically.

* Presentation: The paper is clearly written and well structured

* Significance: The paper addresses an important challenge in differentiable optimization related to its scalability as the size of the problem increases which results in expensive computation of Hessian when using existing methods. Differentiable optimization being a corner stone of modern machine learning algorithm for data science, the paper and the available package clearly provide  practical utility.

* Originality: The paper introduces a new point of view  on differentiable optimization through the lens of bi-level optimization with contributions that are clearly distinguished from existing works. A careful comparison with existing methods through extensive numerical experiments is also a valuable contribution.

---

> ### Author Rebuttal · Authors · 2026-03-31
>
> We sincerely thank Reviewer RMnH for the positive assessment and supportive comments. We are glad that the reviewer found our contributions clear and the experimental results convincing!

---

> > ### Author Rebuttal · Reviewer_RMnH · 2026-04-01
> >
> > I acknowledge the response by the authors. Since there were no questions to discuss, I keep my overall recommendation.

---

> > > ### Author Response · Authors · 2026-04-06
> > >
> > > We thank the reviewer for acknowledging our response and for the positive evaluation of our work.

---

### Official Review · Reviewer_UrZN · 2026-03-12

**Soundness:** 3
**Presentation:** 3
**Significance:** 2
**Originality:** 2
**Overall Recommendation:** 4
**Confidence:** 3

**Summary:**

This paper proposes a fully first-order method for computing gradients through differentiable optimization layers. The key idea is to reformulate the differentiable optimization problem as a bilevel optimization problem, then apply an active-set reduction that transforms general convex constraints into linear equality constraints. This enables the use of a finite-difference hypergradient oracle from Kornowski et al. (2024) to compute approximate hypergradients in $O(1)$ time, yielding an overall complexity of $O(\delta^{-1}\epsilon^{-3})$ for constrained bilevel optimization. The authors release an open-source PyTorch library with a drop-in replacement interface for CvxpyLayer and experiment on two problems.

**Compliance With Llm Reviewing Policy:**

Affirmed.

**Final Justification:**

The author addressed my concern.

**Key Questions For Authors:**

- Figure 11 shows the gradient approximation error, but does not provide the problem considered. I wonder what kinds of problems the method gives a good approximation for and what scenarios it struggles with.
- How does the method perform compared with Cvxpy on SDP?
- How does the method scale with the number of constraints? The theory suggests no dependence, but practical solvers may struggle.

**Limitations:**

yes

**Strengths And Weaknesses:**

**Strength**:
- The problem is well-motivated. The computational bottleneck of Hessian inversion is a genuine practical concern in differentiable optimization layers.
- Elegant method with practical implementation. By leveraging active-set reduction, this paper extends the method in [1] to the inequality-constrained case, making general inequality-constrained differentiable optimization layers more applicable in practice.
- Promising experimental results. In the experiment (although very limited), the proposed method achieves a much faster speed than Cvxpylayer with comparable performance.

**Weekness**:
- Limited novelty relative to [1]. The core theoretical machinery — the finite-difference hypergradient oracle, the nonsmooth nonconvex meta-algorithm, the online-to-nonconvex reduction framework — is entirely borrowed from [1]. The main theoretical contribution here is the active-set reduction. While this is a useful observation, the proof of Theorem 4.5 is straightforward.
- Week Experiment evaluation. The experiments are limited to three tasks (synthetic DFL, Sudoku, and SOCP), all relatively small-scale. For a paper motivated by scalability concerns, there is no experiment demonstrating behavior at scale. Also, although the experiment involves QP and SOCP problems, performance on more general problems (e.g. SDP) are not tested.
- Missing discussion of failure modes. The motivating example in equation (30) shows that general convex constraints can produce non-Lipschitz value functions. The paper introduces Assumption 4.6 to rule this out but does not discuss how to verify these assumptions in practice or what happens when they are violated. A practitioner using the library has no way to know whether the theoretical guarantees apply to their problem.

[1] Kornowski, G., Padmanabhan, S., Wang, K., Zhang, Z., and Sra, S. First-order methods for linearly constrained bilevel optimization. In The Thirty-eighth Annual Conference on Neural Information Processing Systems (NeurIPS), 2024.

---

> ### Author Rebuttal · Authors · 2026-03-31
>
> We thank the reviewer for thoughtful and constructive feedback and comments. Below, we provide a point-by-point response to reviewer UrZN's review.
>
> ---
> **Response to weaknesses**
>
> >Limited novelty relative to [1]. The core theoretical machinery ... is entirely borrowed from [1].
>
> Despite the simplicity of the proof of Theorem 4.5, it has several important theoretical contributions. First, we prove the convergence rate matches the **optimal rate** for nonsmooth nonconvex optimization (Zhang et al., 2020), whereas the convergence rate in [1] is suboptimal. Second, unlike [1], considering only linear constraints, our theorem works for **general convex constraints**, which was highlighted as an open problem in [1].
>
> Beyond theory, our proposed FFOLayer is a complete, open-source solver that serves as a drop-in replacement for CvxpyLayer. The **solver-agnostic** design (Section 5) and the engineering required to make this practical are nontrivial contributions that [1] does not address.
>
> >Week Experiment evaluation. The experiments are limited to three tasks (synthetic DFL, Sudoku, and SOCP), all relatively small-scale.
>
> We want to claim that the experiment scale in this paper is consistent with the standard domains commonly considered in prior differentiable optimization solver papers. For example, in OptNet and LPGD, the largest input size is **729**; and in BPQP, the largest problem dimension is **500**. In our experiment, the largest input dimension is **1000**. Thus, the problem sizes we study are representative of the scales in the differentiable optimization domain.
>
> Moreover, our scalability experiments (Figures 5 and 6) demonstrate the key practical advantage: near-constant memory and sublinear time growth, while CvxpyLayer and LPGD run out of memory at $d_y = 1000$. These scaling trends are arguably more informative than additional tasks at a small scale.
>
> >Missing discussion of failure modes ...The paper introduces Assumption 4.6 to rule this out, but does not discuss how to verify these assumptions in practice or what happens when they are violated.
>
> Thank you for your comment. We've added a runtime error message that flags when the optimality gap between the original and ghost problem is large.
>
> ---
> **Response to key questions**
>
> >Q1: Figure 11 shows the gradient approximation error, but does not provide the problem considered. I wonder what kinds of problems the method gives a good approximation for and what scenarios it struggles with.
>
> We apologize for the oversight. The problem considered in Figure 11 is the QP problem for the synthetic DFL task (Section 6.1). We will fix this in the final draft.
>
> Our method works well for strongly convex lower-level problems. In this case, the finite-difference approximation is accurate (as in our QP and SOCP experiments). Our method may struggle with non-strongly convex objectives or problems near degeneracy, where the active set can change frequently with small perturbations of $x$.
>
> >Q2: How does the method perform compared with Cvxpy on SDP?
>
> Given that most of the solvers (except for CvxpyLayer) do not support SDP, we primarily test on QP and SOCP for the sake of comparison on the computation cost and convergence. We do want to clarify that our current implementation also **supports SDP** (please check the code appendix attached with the submission). We have additionally added a test case for SDP, and preliminary results show that FFOLayer can successfully differentiate through SDP problems with accurate gradients. We plan to include a more thorough SDP evaluation in an extended version.
>
> Please see our updated code here for the new SDP test case: https://anonymous.4open.science/r/FFOLayer-B78B/README.md.
>
> >Q3: How does the method scale with the number of constraints? The theory suggests no dependence, but practical solvers may struggle.
>
> This is an insightful question. To focus more on the dependency on the stationarity gap, we omit the dependency on the number of constraints in our draft. To answer the reviewer's question, each call to the lower-level solver has a cost that depends on the problem structure, including the number of constraints.
>
> In practice, the number of constraints affects our method in two ways. First, for the forward cost, more constraints increase the cost of each lower-level solve, but this cost is shared by all methods (including CvxpyLayer). Second, our backward cost depends mainly on the **number of active constraints**, not the total number. For well-structured problems (e.g., sparse QPs), the active set is typically much smaller than the total constraint count, giving our method an additional advantage.
>
> ---
> We hope these clarifications and additional results resolve all questions and outstanding concerns. We are happy to answer any remaining questions. Thank you for your detailed and thoughtful review.

---

> > ### Author Rebuttal · Reviewer_UrZN · 2026-04-02
> >
> > Thanks for the author's response. I tested the SDP case, and it provides good approximations. Together with the results in the paper, I think this paper is of good practical value. I increased the score.

---

> > > ### Author Response · Authors · 2026-04-06
> > >
> > > We sincerely thank the reviewer for testing our method on the SDP case and for the positive feedback. We are glad that the results demonstrate the practical value of our approach!

---

### Official Review · Reviewer_z97v · 2026-03-13

**Soundness:** 3
**Presentation:** 1
**Significance:** 3
**Originality:** 3
**Overall Recommendation:** 4
**Confidence:** 3

**Summary:**

The paper studies the problem of differentiating through an optimization problem with inequality constraints that can be used as a layer in neural networks. The authors convert the inequality-constrained problem into an equality-constrained formulation and leverage the analysis from a previous work to compute the hypergradient using a finite-difference approach. This avoids the need to compute second-order derivatives of the lower-level objective and to invert its Hessian, which are typically required in implicit differentiation. The proposed method is compared empirically with several existing approaches that rely on constructing and inverting the KKT matrix of the problem.

**Compliance With Llm Reviewing Policy:**

Affirmed.

**Final Justification:**

I am satisfied by the authors' response and therefore have changed my score from weak reject to weak accept.

**Key Questions For Authors:**

I have a few questions:
1. On Line 161, it is stated that "... involves computing a Hessian ..., which poses a major bottleneck for scaling differentiable optimization layers". Could the authors clarify why Hessian computation becomes the main bottleneck in hypergradient computation? In many modern implementations, vector-Hessian products can be computed efficiently without forming the Hessian explicitly.
2. Replacing the Hessian inversion in implicit differentiation by finite differences requires solving a second optimization problem (P3) alongside (P0). Could the authors comment on how difficult it is to solve this additional problem in practice? In particular, can the solver for this problem be warm-started using the solution of (P0)? Otherwise it seems that one bottleneck may simply be replaced with another.
3. Implicit differentiation methods often benefit significantly from warm-starting (see [1,2]). How does the proposed approach compare to previous techniques if those methods are also warm-started?

**References**

[1] Ji, K., Yang, J. and Liang, Y., 2021, July. Bilevel optimization: Convergence analysis and enhanced design.

[2] Arbel, M. and Mairal, J., 2021. Amortized implicit differentiation for stochastic bilevel optimization.

**Limitations:**

Not applicable.

**Strengths And Weaknesses:**

**Strengths**
1. Although the paper leverages heavily on a previous work, their contributions are original.
2. The authors cleverly use the work of [1] on equality constrained problem, convert the inequality constrained problem to equality constrained and then simply outperform their work on inequality constrained, which is impressive.
3. The presented approach avoids the need to compute the second order derivatives altogether.
4. The experimental results suggest improved scalability compared to several existing differentiable optimization layers.

**Weaknesses**
1. While the overall idea is interesting, several parts of the paper are difficult to follow and some design choices are insufficiently motivated.
2. The first line in the first stated contribution, corresponding to the reformulation from (P0) to (P1), appears to be a straightforward observation. It may be more helpful if the paper instead focuses on clearly motivating the subsequent step that converts the inequality-constrained problem into an equality-constrained formulation and leverages the results of [1].
3. The transition from Problem (P1) to Problem (P2) is not entirely clear. In particular, the role of the reference point $\bar x$ and how the resulting equality constraints relate to the original inequality constraints could be explained more carefully.
4. In Section 5, the authors construct an alternative problem using the detached gradient of the upper objective with respect to $y$. The motivation for this step is not clearly explained. The authors should clarify (in the main text) a bit more why using the detached gradient (as compared to directly using the upper objective $f(x,y)$) has any significance?
5. The proposed strategy requires solving two optimization problems instead of one in the forward pass.

**References**

[1] Kornowski, G., Padmanabhan, S., Wang, K., Zhang, Z., and Sra, S., 2024, First-order methods for linearly constrained bilevel optimization.

---

> ### Author Rebuttal · Authors · 2026-03-31
>
> We thank the reviewer for thoughtful and constructive feedback and comments. Below, we provide a point-by-point response to reviewer z97v's review.
>
> ---
> **Response to weaknesses**
>
> >The first line in the first stated contribution, corresponding to the reformulation from (P0) to (P1), appears to be a straightforward observation.
>
> We appreciate this observation. From Problem P0 to P1, although it sounds trivial, it reframes the problem as bilevel rather than single-level (e.g., Cvxpylayer, qpth, etc), enabling fully first-order bilevel methods.
>
> >The transition from Problem (P1) to Problem (P2) is not entirely clear. In particular, the role of the reference point $\bar{x}$ and how the resulting equality constraints relate to the original inequality constraints could be explained more carefully.
>
> Thank you for highlighting this. The transition from inequality-constrained P1 to equality-constrained (P1) is just building the tangent space of the active constraints. Under a locally fixed active set, first-order changes in $y^*(x)$ are governed by the tangent space at $(\bar{x},y)$ (as Figure 1 illustrates), so P2 has the same local derivative as P1 (Theorem 4.5).
>
> >In Section 5, ... The authors should clarify (in the main text) a bit more why using the detached gradient (as compared to directly using the upper objective $f(x, y))$ has any significance?
>
> We replace the upper objective $f(x, y))$ with a new surrogate $f(x, y))$ mainly because of practice consideration. As written in Lines 270-274, our goal is to make FFOLayer solver-agnostic: the user specifies only the lower-level problem, while the solver should work as a black box without depending on the explicit form of the downstream loss $f$. In other words, we need to optimize $g + \delta f$ even if we **don't know** the explicit form of $f$.
>
> > The proposed strategy requires solving two optimization problems instead of one in the forward pass.
>
> Thank you for your concern. We agree that our method requires solving two optimization problems. That said, we would like to emphasize that the second problem is much easier to solve in practice because it can be warm-started using the solution of the forward pass. In addition, unlike in implicit differentiation, our method doesn't require solving the linear system via conjugate gradient.
>
> ---
> **Response to key questions**
>
> >Q1: On Line 161... Could the authors clarify why Hessian computation becomes the main bottleneck in hypergradient computation? In many modern implementations, vector-Hessian products can be computed efficiently without forming the Hessian explicitly.
>
> Thank you for the question. Using Hessian-vector products is a standard and widely adopted approach in implicit differentiation methods (e.g., used in CvxpyLayers). We've already compared this in our experiments.
>
> We want to emphasize that although vector-Hessian products can avoid explicit construction of the Hessian, its overall cost depends heavily on the condition number of the KKT system, and it largely relies on how conjugate gradient works. As a result, the computation cost can become substantial when the KKT system is ill-conditioned. In contrast, our fully first-order method is more efficient by bypassing the KKT system and therefore doesn't rely on the conjugate gradient.
>
> >Q2: Replacing the Hessian inversion in implicit differentiation by finite differences requires solving a second optimization problem (P3) alongside (P0). Could the authors comment on how difficult it is to solve this additional problem in practice?
>
> Please see our response above for "The proposed strategy requires solving two optimization problems instead of one in the forward pass."
>
> >Q3: In particular, can the solver for this problem be warm-started using the solution of (P0)? ... How does the proposed approach compare to previous techniques if those methods are also warm-started?
>
> Thank you for the question. In general, the solution of the forward pass (P0) cannot be used to warm-start the backward pass in standard implicit differentiation. The reason is that the backward pass is not solving for the primal optimum again; it is computing the implicit derivative.  There is no known general mapping or heuristic that turns the solution of P0 into a good initialization for the derivative computation.
>
> However, in our method, since we don't rely on implicit differentiation, our method can naturally take advantage of warm-starting. We also conduct an additional Sudoku experiment, which shows that we can reduce the backward time from **140** seconds to **65** seconds by warm-starting.  Please see our updated code here (comp_warm_start.ipynb): https://anonymous.4open.science/r/FFOLayer-B78B/README.md.
>
> ---
> We hope these clarifications and additional results resolve all questions and outstanding concerns. We are happy to answer any remaining questions. Thank you for your detailed and thoughtful review.

---

> > ### Author Rebuttal · Reviewer_z97v · 2026-04-04
> >
> > Thank you for the detailed rebuttal and clarifications. Some of my concerns were addressed, but several questions remain.
> >
> > > "... From Problem P0 to P1, although it sounds trivial, it reframes the problem ..."
> >
> > My concern was that this step is not only straightforward but also widely known. In bilevel optimization, implicit models are commonly presented as standard examples precisely due to this observation. Therefore, presenting the reformulation from (P0) to (P1) as a contribution appears somewhat misleading.
> >
> > > "... the second problem is much easier to solve in practice because it can be warm-started using the solution ..."
> >
> > I did not find this discussed explicitly in the paper. If warm-starting indeed makes the second problem easier to solve, it would be helpful for the authors to clearly mention this in the manuscript and, if possible, demonstrate it empirically.
> >
> > > "... its overall cost depends heavily on the condition number of the KKT system ..."
> >
> > Does the condition number of the KKT matrix not also influence the convergence speed of the proposed algorithm? If so, it would be helpful to clarify how the conditioning affects both the proposed approach and the baseline methods.
> >
> > > "... the solution of the forward pass (P0) cannot be used to warm-start the backward pass ..."
> >
> > I think the authors misunderstood my question. In Algorithm 1, can solving the lower problem in (P3) for the primal–dual pair $(y_{\delta}^\*, \lambda_{\delta}^\*)$ in Step 3 be warm-started using the primal–dual pair $(y^\*, \lambda^\*)$ obtained from solving (P0) in Step 2? The authors answered that in response to my another question that the Step 3 can indeed be warm-started. However, the authors do not mention it in the paper.
> >
> > Furthermore, in implicit differentiation, the backward pass involves solving a linear system whose adjoint solution can often be stored and used to warm-start subsequent outer iterations. How does the proposed method compare to warm-started implicit differentiation in this setting?
> >
> > Overall, while the rebuttal clarified several points, these questions still influence my assessment.

---

> > > ### Author Response · Authors · 2026-04-06
> > >
> > > We thank Reviewer z97v for engaging with the rebuttal and providing their valuable feedback. We are delighted that we could clarify several of their concerns and are truly grateful for their decision to bring this work forward. We are thankful to be provided with an opportunity to address their pending concerns and shall answer them as follows.
> > >
> > > >(P0) to (P1) is not only straightforward but also widely known.
> > >
> > > We thank the reviewer for this comment. The reformulation from (P0) to (P1) is included mainly for completeness of the proof and for people who are not familiar with bilevel optimization. We will ensure the related work of this reduction is clearly summarized in the revised paper to avoid any potential misunderstanding.
> > >
> > > >Does the condition number of the KKT matrix not also influence the convergence speed of the proposed algorithm?
> > >
> > > Our approach doesn't rely on differentiating through the KKT system so it avoids dependence on the **condition number $\kappa$ of KKT system**. Our method instead relies on solving a Lagrangian minimization in the backward pass. Solving the Lagrangian minimization can benefit from warm starting the solution by the forward pass and can be solved by any optimization solver (solver-agnostic), while differentiating through the KKT system needs to rely on iterative linear solvers such as conjugate gradient and may not directly leverage off-the-shelf optimization solvers.
> > >
> > > On the other hand, if the Lagrangian minimization in Step 3 is solved from scratch, then the complexity in Theorem 4.6 indeed has a dependency on a **different condition number** $\kappa' := L/\mu$ (smoothness over strong convexity of the lower-level function) and makes the complete complexity $O(\sqrt{\kappa'} \log{\epsilon})$. This condition number is the same as the top-left block of the KKT matrix and thus is generally smaller than the condition number of the full KKT matrix. This reduction in the condition number improves the convergence rate of our algorithm. In practice, the Lagrangian minimization in Step 3 can also be warm-started from the forward pass and lead to a further speedup.
> > >
> > > > In Algorithm 1, can solving the lower problem in (P3) for the primal–dual pair in Step 3 be warm-started using the primal–dual pair obtained from solving (P0) in Step 2?
> > >
> > > Yes. As our new added experiment result shows, we can reduce **over 50% of the total solving time** for the lower problem in (P3) by warm-starting from the solution of (P0).
> > >
> > > The discussion for warm-start and condition number are not included in our current version. We will revise our manuscript in the final version to include this.
> > >
> > > > Furthermore, in implicit differentiation, the backward pass involves solving a linear system whose adjoint solution can often be stored and used to warm-start subsequent outer iterations. How does the proposed method compare to warm-started implicit differentiation in this setting?
> > >
> > > We thank the reviewer for this suggestion. We compare the two warm-start strategies below.
> > >
> > > For implicit differetiationm the backward pass solves a linear system, and one can store the adjoint solution to warm-start the next iteration. However, in constrained settings, when the active set of the lower-level constraint **changes between iterations**, the adjoint system is defined by a different system, and thus the adjoint solution from the previous iteration may no longer be a good initial guess. In addition, for methods like CvxpyLayer, the canonicalization to cone programs introduces further **shifts** in the adjoint solution space, making cross-iteration warm-starting less effective.
> > >
> > > In contrast, our method warm-starts the backward pass (Step 3) from the forward pass (Step 2) **within the same iteration**. By the proof of our Theorem 4.6, the two solutions of (P3) and (P0) are close up to $O(\delta)$, where $\delta$ is the solver tolerance, providing a stronger warm-start guarantee that is not affected by active set changes across iterations. Thus our warm start is expected to perform better. This is also validated in our new experiment, where we see an over 50% reduction in the computation cost of the backward pass while using warm start.

---

### Official Review · Reviewer_WQxa · 2026-03-13

**Soundness:** 3
**Presentation:** 3
**Significance:** 3
**Originality:** 3
**Overall Recommendation:** 5
**Confidence:** 3

**Summary:**

This paper is motivated by the high cost of existing differentiable optimization layers, which usually depend on KKT-based implicit differentiation and second-order computations. To address this, it proposes FFOLayer, a fully first-order, solver-agnostic differentiable optimization layer that can efficiently handle general convex constraints while offering theoretical guarantees and practical drop-in usability.

**Compliance With Llm Reviewing Policy:**

Affirmed.

**Final Justification:**

I think my concern and question are answered. I will keep my score.

**Key Questions For Authors:**

I am curious about the extent to which the choice of solver affects convergence performance. In addition, it would be valuable to understand whether the proposed method can scale to larger and more practical problem settings.

**Limitations:**

yes

**Strengths And Weaknesses:**

Strengths:
1. The paper addresses a real bottleneck in differentiable optimization layers, where KKT-based implicit differentiation is often computationally and memory-expensive.
2. FFOLayer is fully first-order, supports general convex constraints, and is solver-agnostic, which is a strong and meaningful combination.
3. The work provides theoretical guarantees and also offers a practical drop-in replacement for existing differentiable optimization layers.

Weaknesses:
1. The method appears to rely on rather strong assumptions, including upper-level smoothness/Lipschitz continuity and lower-level smoothness/strong convexity. While these assumptions are useful for the theoretical analysis, they may limit the practical generality of the approach and make it less clear how well the method extends to more realistic or less well-structured settings.
2. The experimental evaluation is somewhat limited and not fully convincing. For instance, the convergence curves in Figure 3 are uniformly fast and highly similar across methods, which makes the comparison less informative. More broadly, the experimental setups seem relatively simple and do not sufficiently demonstrate the effectiveness of the proposed method in realistic machine learning scenarios.

---

> ### Author Rebuttal · Authors · 2026-03-31
>
> We thank the reviewer for thoughtful and constructive feedback and comments. Below, we provide a point-by-point response to reviewer WQxa's review.
>
> >Q1: Rely on strong assumptions.
>
> We appreciate the concern regarding our assumptions. Assumptions 4.2 (smoothness and strong convexity of the lower level) and 4.3 (active-set identification) are standard in the bilevel optimization literature. For instance, smoothness and strong convexity of the lower-level objective are assumed in nearly all prior work on first-order bilevel methods (Kwon et al., 2023; 2024a; Kornowski et al., 2024). The active-set identification assumption is similarly common for constrained bilevel problems (Khanduri et al., 2025) and holds generically for strongly convex lower-level problems with linear constraints.
>
> > Q2: The convergence curves in Figure 3 are highly similar across methods; the experimental setups seem relatively simple and do not sufficiently demonstrate the effectiveness of the proposed method in realistic machine learning scenarios.
>
> This similarity is actually expected and validate our approach. Except for LPGD and our methods (FFOCP and FFOQP), all other methods are exact-gradient methods, meaning that they should have the same convergence performance. For approximated-gradient methods, our methods have comparable convergence to the exact-gradient methods but LPGD suffers.
>
> >Q3: I am curious about the extent to which the choice of solver affects convergence performance.
>
> As stated in Line 292, we use SCS as the default solver because 1) a fair comparion to the baselines like CvxpyLayer; 2) it is publicly available. Importantly, our implementation supports different choices of the solver and it is an option to specify in the library. Empirically, we do find that using advanced solvers like GUROBI can improve the computation cost in certain problems.
>
>
> >Q4: In addition, it would be valuable to understand whether the proposed method can scale to larger and more practical problem settings.
>
> We want to claim that the experiment scale in this paper is consistent with the standard domains commonly considered in prior differentiable optimization solver papers. For example, in OptNet and LPGD, the largest input size is **729**; and in BPQP, the largest problem dimension is **500**. In our experiment, the largest input dimension is **1000**. Thus, the problem sizes we study are representative of the scales in the differentiable optimization domain.
>
> Moreover, our scalability experiments (Figure 5 and 6.) demonstrate the key practical advantage: near-constant memory and sublinear time growth, while CvxpyLayer and LPGD run out of memory at $d_y = 1000$. These scaling trends are arguably more informative than additional tasks at a small scale.
>
> ---
>
>
> We hope these clarifications and additional results resolve all questions and outstanding concerns. We are happy to answer any remaining questions. Thank you for your detailed and thoughtful review.

---

> > ### Author Rebuttal · Reviewer_WQxa · 2026-04-05
> >
> > I think my concern and question are answered. I will keep my score.

---

> > > ### Author Response · Authors · 2026-04-06
> > >
> > > We thank the reviewer for the constructive feedback, which helped improve our paper.

---

### Decision · Program_Chairs · 2026-04-30

**Decision:**

Accept (spotlight)

**Comment:**

This paper introduces FFOLayer, a fully first-order, solver-agnostic differentiable optimization layer that reformulates KKT differentiation as bilevel optimization to compute approximate hypergradients using only near-constant first-order information, which leverages the recent advances of bilevel optimization with constraints (e.g., Shen and Chen, 2023, Kwon et al., 2023, Kornowski et al. 2024). The paper provides theoretical convergence guarantees and releases an open-source drop-in replacement for CvxpyLayer.

Reviewers are in broad agreement that the paper addresses a genuine practical bottleneck in differentiable optimization. The primary criticisms centered on two issues. Reviewer UrZN noted that the core theoretical machinery is borrowed from prior work on first-order methods for bilevel optimization, with the active-set reduction being a straightforward extension. Multiple reviewers noted limited experimental scope.

The authors provided a strong rebuttal to the above concerns. A new SDP experiment was added and independently verified by Reviewer UrZN. After the rebuttal, all four reviewers indicated their concerns were either fully or partially resolved. I am also convinced that the paper makes a solid technical contribution with clear practical impact. Therefore, I recommend Accept.